# Gaussian synapses for probabilistic neural networks

Amritanand Sebastian [1], Andrew Pannone[1], Shiva Subbulakshmi Radhakrishnan [1,2] & Saptarshi Das [1,3,4]

The recent decline in energy, size and complexity scaling of traditional von Neumann architecture has resurrected considerable interest in brain-inspired computing. Artificial neural networks (ANNs) based on emerging devices, such as memristors, achieve brain-like computing but lack energy-efficiency. Furthermore, slow learning, incremental adaptation, and false convergence are unresolved challenges for ANNs. In this article we, therefore, introduce Gaussian synapses based on heterostructures of atomically thin two-dimensional (2D) layered materials, namely molybdenum disulfide and black phosphorus field effect transistors (FETs), as a class of analog and probabilistic computational primitives for hardware implementation of statistical neural networks. We also demonstrate complete tunability of amplitude, mean and standard deviation of the Gaussian synapse via threshold engineering in dual gated molybdenum disulfide and black phosphorus FETs. Finally, we show simulation results for classification of brainwaves using Gaussian synapse based probabilistic neural networks.

[1] Department of Engineering Science and Mechanics, Pennsylvania State University, University Park, PA 16802, USA. [2] Electrical and Electronics Engineering, Amrita Vishwa Vidyapeetham, Ettimadai, Coimbatore, Tamil Nadu 641112, India. [3] Department of Materials Science and Engineering, Pennsylvania State University, University Park, PA 16802, USA. [4] Materials Research Institute, Pennsylvania State University, University Park, PA 16802, USA. Correspondence and requests for materials should be addressed to S.D. (email: sud70@psu.edu)

The last five decades have witnessed an unprecedented and exponential growth in computational power, primarily driven by the success of the semiconductor industry. Relentless scaling[1] of complementary metal oxide semiconductor (CMOS) technology enabled by breakthroughs in material discovery[2], innovation in device physics[3], transformation in micro and nanolithography techniques[4], and the triumph of von Neumann architecture[5] contributed to the computing revolution. Scaling has three characteristic aspects to it. Energy scaling to ensure practically constant computational power budget, size scaling to ensure faster and cheaper computing since more and more transistors can be packed into the same chip area, and complexity scaling to ensure incessant growth in computational power of single on-chip processor. The golden era of metal oxide semiconductor field effect transistor (MOSFET) scaling, also referred to as the Dennard scaling era[6], has witnessed concurrent scaling of all three aspects for almost three decades. However, around 2005, the energy scaling ended owing to fundamental thermodynamic limitations at the device physics level, popularly known as the Boltzmann tyranny[7]. Size scaling continued for another decade albeit with new challenges[8] and eventually ended in 2017 owing to limitations at the materials level imposed by quantum mechanics[1]. Complexity scaling is also on decline owing to the non-scalability of traditional von Neumann computing architecture and the impending "Dark Silicon" era that presents a severe threat to multi-core processor technology[9]. In order to sustain the growth in computational power, it is imperative that all three aspects of scaling must be reinstated immediately through material rediscovery, device innovations, and advancement in higher complexity computing architectures.

The extraordinarily complex neurobiological architecture of the mammalian nervous system that seamlessly executes diverse and intricate functionalities such as adaption, perception, acquisition of sensory information, learning, memory formation, emotions, cognition, motor action, and many more has inspired computer scientists to think beyond the traditional von Neumann architecture in order to resurrect complexity scaling. The neural architecture deploys billions of information processing units, neurons, which are connected via trillions of synapses in order to accomplish massively parallel, synchronous, coherent, and concurrent computation. This is markedly different from the von Neumann architecture, where the logic and memory units are physically isolated and operate sequentially, i.e., instruction fetch and data operation cannot occur simultaneously. Furthermore, unlike the deterministic digital switches (logic transistors), neural architecture uses probabilistic and analog computational primitives in order to accomplish adaptive functionalities such as pattern recognition and pattern classification, which form the foundation for mammalian problem solving and decision making.

In the above context, IBM's bioinspired CMOS chip, True North, is a remarkable breakthrough in neuromorphic computing, achieving the complexity of more than 1 million neurons or 256 million synapses while consuming a miniscule 70 mW of power[10]. Similarly, extensive work by Luca Benini et al. have recently shown that hardware digital neural networks consume comparable or less energy than the human brain for complex tasks, such as, image recognition[11]. While these are impressive advancements, the inherent scaling challenges associated with the digital CMOS technology can ultimately limit the implementation of very-large-scale artificial neural networks (ANNs), invoking the critical and imminent need for energy efficient analog computing primitives for ANNs. Recent years have, therefore, witnessed innovation in analog devices such as the memristors[12–14], coupled oscillators[15], and various targeted components[16–18], which can emulate neural spiking, neural transmission, and neural plasticity and hence can be used as computational

primitives in ANNs. While these devices do provide some energy benefit at the architectural level for specific applications, they fail to address the intrinsic energy and size scaling needs at the device level, which can ultimately lead to stagnation in complexity scaling. Further challenges associated with ANNs are often overlooked. For example, ANNs deployed for pattern classification problems require optimum training algorithms and learning rules to identify the class statistics with desired accuracy within a short training time. Unfortunately, the most popular and widely used heuristic backpropagation algorithm[19], is inherently slow and remains vulnerable to local minima in spite of extensive modifications in recent years using methods such as conjugate gradient, quasi-Newton, and Levenberg–Marquardt (LM) to improve the convergence[20]. In order to address the slow learning, incremental adaptation, and inherent unreliability of ANNs, novel classification techniques based on statistical principles must be embraced.

In this article, we experimentally demonstrate how a new class of analog devices, namely, Gaussian synapses, based on the heterostructure of novel atomically thin two dimensional (2D) layered semiconductors enables the hardware implementations of probabilistic neural networks (PNNs) and thereby reinstates all three aforementioned quintessential scaling aspects of computing. In short, 2D materials facilitate aggressive size scaling, analog Gaussian synapses offer energy scaling, and PNNs enable complexity scaling. Combined, these new developments can facilitate *Exascale Computing* and ultimately benefit scientific discovery, national security, energy security, economic security, infrastructure development, and advanced healthcare programs[21,22].

## Results

**Probabilistic neural network**. Our overall approach is summarized in Fig. 1a. First, we reintroduce PNN that was proposed by Specht, D. F.[23] PNN is derived from Bayesian computing and Kernel density method[24]. As shown in Fig. 1b, unlike ANNs, which necessitate multiple hidden layers, each with a large number of nodes, PNNs are comprised of a pattern layer and a summation layer and can map any input pattern to any number of output classifications. Furthermore, ANNs use activation functions such as sigmoid, rectified linear unit (ReLU), and their various derivatives for determining pattern statistics. This is often extremely difficult to accomplish with reasonable accuracy for non-linear decision boundaries. PNNs, on the contrary, use parent probability distribution functions (PDFs) approximated by Parzen window[25] and a non-parametric function to define the class probability, which in the case of Gaussian kernel is defined by Gaussian distribution as shown in Fig. 1c. PNNs, therefore, facilitate seamless and accurate classification of complex patterns with arbitrarily shaped decision boundaries. Furthermore, PNNs can be extended to map higher dimensional functions since multivariate Gaussian kernels are simply the product of univariate kernels as shown in Fig. 1d.

Unfortunately, in spite of the widespread applications and simplicity of PNNs, their hardware implementation is rather sparse. In fact, neither ANNs nor PNNs have been extensively realized using hardware components. While there is growing interest towards the development of devices for the hardware implementation of ANNs, the effort and investment towards the hardware implementation of PNNs are still very limited. One reason is that the hardware implementation of probability functions associated with the PNNs, such as the Gaussian, requires multicomponent digital CMOS circuits that leads to severe area and energy inefficiency. For example, the "Bump" circuit demonstrated by Delbruck, T. uses seven transistors[26]. Similarly, the Gaussian synapse proposed by Choi, J. et al. consists

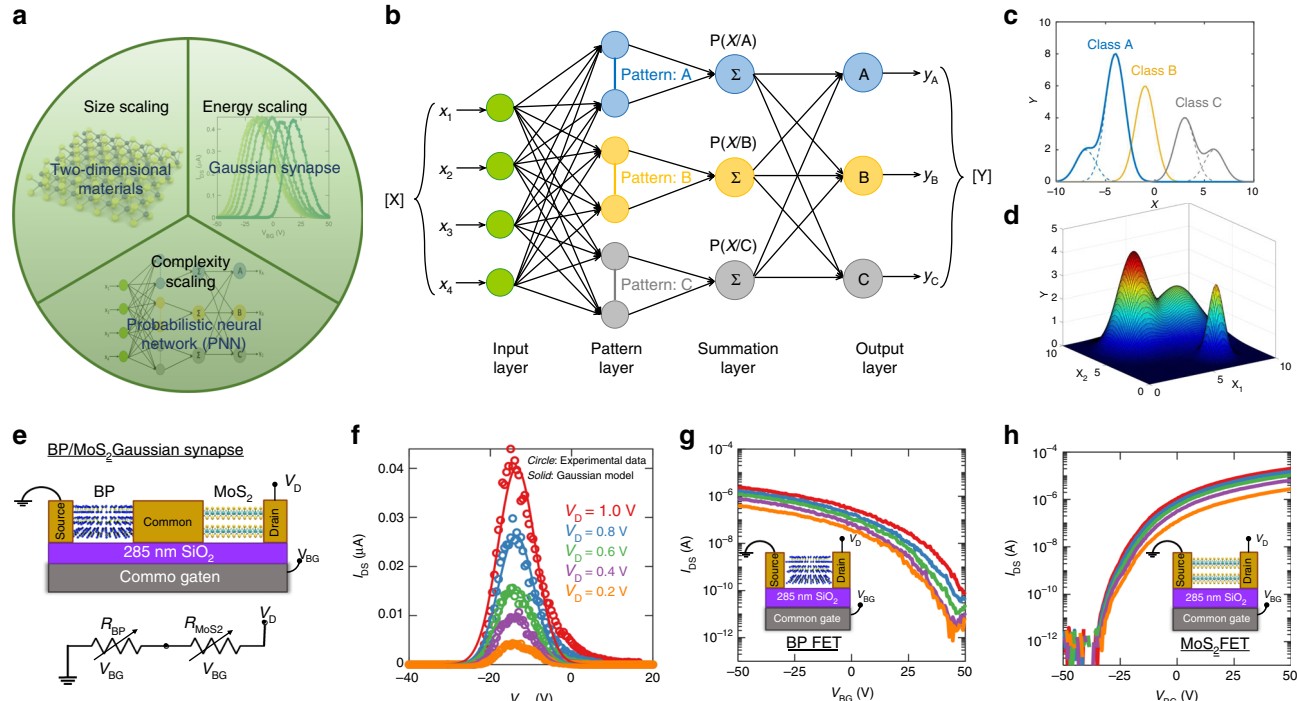

**Fig. 1** Gaussian Synapse based probabilistic neural network (PNN). **a** Resurrection of three quintessential scaling aspects of computation i.e., complexity scaling through PNNs, size scaling through atomically thin 2D materials and energy scaling through analog Gaussian synapses. **b** Schematic representation of PNN that comprise of a pattern layer and a summation layer for mapping any input pattern to any number of output classifications. **c** Gaussian probability density functions (PDFs) facilitating seamless and accurate classification of complex patterns with arbitrarily shaped decision boundaries. **d** Multivariate Gaussian kernel for mapping higher dimensional functions. **e** Schematic of two transistor Gaussian synapse based on heterogeneous integration of n-type $MoS_2$ and p-type black phosphorus (BP) back-gated field effect transistors (FETs). The equivalent circuit diagram consists of two variable resistors connected in series. **f** Transfer characteristics i.e., the drain current ($I_D$) versus back-gate voltage ($V_G$) of the Gaussian synapse for different drain voltages ($V_D$). Clearly, the experimental data (circles) can be modeled by Gaussian distributions (solid). **g** Transfer characteristics of p-type BP FET. **h** Transfer characteristics of n-type $MoS_2$ FET

of a pair of differential amplifiers and several arithmetic computational units[27]. Madrenas, J. et al. introduced an alternate method to obtain Gaussian function by combining the exponential characteristics of MOSFETs in sub-threshold and square characteristics in inversion[28]. The circuit was further improved for better symmetry and greater control and tunability by adding more transistors in a floating gate configuration[29,30]. Another approach is to use a Gilbert Gaussian function, where two sigmoid curves are combined using a differential pair along with a current mirror[31,32]. Without any extra circuitry to reduce asymmetry, the most compact circuit uses five transistors[28]. As we will discuss in the following section, our experimental demonstration of Gaussian synapses uses only two transistors, which significantly improves the area and energy efficiency at the device level and provides cascading benefits at the circuit, architecture, and system levels. This will stimulate the much-needed interest in the hardware implementation of PNNs for a wide range of pattern classification problems.

**Gaussian synapse.** Figure 1e shows the schematic of our proposed two transistor Gaussian synapse based on heterogeneous integration of n-type molybdenum disulfide ($MoS_2$) and p-type black phosphorus (BP) back-gated field effect transistors (FETs). Fig. 1e also shows the equivalent circuit diagram for the Gaussian synapse, which simply consists of two variable resistors in series. The two variable resistors, i.e., $R_{MoS_2}$ and $R_{BP}$ correspond to the $MoS_2$ and BP FETs. Fig. 1f shows the experimentally measured transfer characteristics i.e., the drain current ($I_D$) versus back-gate voltage ($V_{BG}$) of the Gaussian synapse for different drain voltages

($V_D$). The fabrication process flow and electrical measurement setup for Gaussian synapses are described in the experimental method section. Clearly, the transfer characteristics resemble a Gaussian distribution which can be modeled using the following equation.

$$I_D = \frac{A}{\sqrt{2\pi\sigma_V^2}} \exp\left[-\frac{(V_{BG} - \mu_V)^2}{2\sigma_V^2}\right]; A = \beta V_D \quad (1)$$

Where, $A$, $\mu_V$ and $\sigma_V$ are, respectively, the amplitude, mean, and standard deviation of the Gaussian. For a specific $MoS_2$/BP pair, $\mu_V$ and $\sigma_V$ are found to be constants, whereas, $A$ varies linearly with $V_D$. The emergence of Gaussian transfer characteristics can be explained using the experimentally measured transfer characteristics of its constituents, i.e., the $MoS_2$ FET and the BP FET, as shown in Fig. 1g, h, respectively. $MoS_2$ FETs exhibit unipolar n-type characteristics, irrespective of the choice of contact metal, owing to the phenomenon of metal Fermi level pinning close to the conduction band that facilitates easier electron injection, whereas, BP FETs are predominantly p-type with large work function contact metals such as Ni[33–36]. Furthermore, unlike conventional enhancement mode Si FETs used in CMOS circuits, both $MoS_2$ and BP FETs are depletion mode, i.e., they are normally ON without applying any back-gate voltage. Remarkably, this simple difference results in the unique Gaussian transfer characteristics for the $MoS_2$/BP pair in spite of the device structure closely resembling a CMOS logic inverter. From the equivalent circuit diagram, the current ($I_D$) through the

Gaussian synapse can be written as:

$$I_D = \frac{V_D}{R_{MoS_2} + R_{BP}} \quad (2)$$

For extreme $V_{BG}$ values, i.e., large negative (lesser than $-30$ V) and large positive (greater than 30 V), the MoS$_2$ FET and the BP FET are in their respective OFF states, making the corresponding resistances, i.e., $R_{MoS_2}$ and $R_{BP}$ very large (approximately TΩ). This prevents any current conduction between the source and the drain terminal of the Gaussian synapse. However, as the MoS$_2$ FET switches from OFF state to ON state, current conduction begins and increases exponentially with $V_{BG}$ following the subthreshold characteristics and reaches its peak magnitude determined by $V_D$. Beyond this peak, the current starts to decrease exponentially following the subthreshold characteristics of the BP FET. As a result, the series connection of the MoS$_2$ and BP FETs exhibits non-monotonic transfer characteristics with exponential tails that mimics a Gaussian distribution.

It must be noted that the Gaussian synapses do not utilize the ON state FET performance and, therefore, are minimally influenced by the carrier mobility values of the n-type and p-type FETs. Instead, the Gaussian synapse exploits the sub-threshold FET characteristics, where the slope is independent of the carrier mobility of the semiconducting channel material. For symmetric Gaussian synapses, it is therefore more desirable to ensure similar sub-threshold slope (SS) values for the respective FETs than the carrier mobility. Ideally, the SS values for both FETs should be 60 mV decade$^{-1}$. However, presence of a nonzero interface trap capacitance worsens the SS. The SS can be improved by minimizing interface states at the 2D/gate-dielectric interface, as well as by scaling the thickness of the gate dielectric. It is also desirable to have FETs with Ohmic contacts for Gaussian synapses to ensure that the SS is determined by the thermionic emission of carriers in order to reach the minimum theoretical value of 60 mV decade$^{-1}$ at room temperature. For Schottky contact FETs, the SS can be severely degraded due to tunneling of carriers through the Schottky barrier.

While our proof-of-concept demonstration of Gaussian synapses is based on exfoliated MoS$_2$ and BP flakes, it is well known that the micromechanical exfoliation is not a scalable manufacturing process for large-scale integrated circuits. Therefore, hardware implementation of PNNs using Gaussian synapses will necessitate large-area growth of MoS$_2$ and BP. Fortunately, recent years have seen tremendous progress in wafer-scale growth of high quality MoS$_2$ and BP using chemical vapor deposition (CVD) and metal organic chemical vapor deposition (MOCVD) techniques[37–41]. Furthermore, while we have used two different 2D materials, MoS$_2$ and BP, for our demonstration of Gaussian synapses owing to their superior performance as n-type and p-type FETs, respectively, there are 2D materials, such as, WSe$_2$, which offer ambipolar transport, i.e., the presence of both electron and hole conduction[42] and can be grown over large area using CVD techniques[43]. However, the performance of WSe$_2$ based n-type and p-type FETs are limited by the presence of large Schottky barriers at the metal/2D contact interfaces[36]. By resolving the contact resistance related issues[36] and improving the quality of large-area synthesized WSe$_2$, it is possible to implement Gaussian synapses based solely on WSe$_2$ as well. Moreover, in recent years several groups have demonstrated p-type MoS$_2$ and n-type BP, by implementing smart contact engineering and/or doping strategies[44,45]. Therefore, very-large-scale integration of Gaussian synapses based on a CVD grown single 2D material will be possible in the near future for the hardware realization of PNNs. Since the focus of this article is to introduce the novel Gaussian synapse and its benefit as a statistical computing primitive, we avoided material optimization.

Gaussian synapses are inherently low power since they exploit the subthreshold characteristics of the FET devices. In this context, we would like to remind the readers that the total power consumption ($P_{total}$) in digital CMOS circuit comprises, primarily, of dynamic switching power ($P_{dynamic}$) and static leakage power ($P_{static}$) and is given by the following equation:

$$P_{total} = P_{dynamic} + P_{static} = \eta C V_{DD}^2 f + I_{static} V_{DD} \quad (3)$$

Note that, $\eta$ is the activity factor, $C$ is the capacitance of the circuit, $f$ is the switching frequency, and $V_{DD}$ is the supply voltage. During the Dennard scaling era, the power consumption of the chip was dominated by $P_{dynamic}$, which was kept constant by scaling the threshold voltage ($V_{TH}$) and concurrently the supply voltage ($V_{DD}$) of the MOSFET. However, beyond 2005, the voltage scaling stalled since further reduction in $V_{TH}$ resulted in an exponential increase in the static leakage current ($I_{static}$) and hence static power consumption. This is a direct consequence of the non-scalability of the subthreshold swing (SS) to below 60 mV decade$^{-1}$, as determined by the Boltzmann statistics. In fact, $P_{total}$ in the present Dark-Si era is mostly dominated by $P_{static}$. Regardless of whether the dynamic or static power dominates, reinstating $V_{DD}$ scaling is the only way to escape the Boltzmann tyranny. This is why in recent years, subthreshold logic circuits, which utilize $V_{DD}$ that is close to or even less than $V_{TH}$, have received significant attention for ultra-low power applications[46,47]. New subthreshold logic and memory design methodologies have already been developed and demonstrated on a fast Fourier transform (FFT) processor[48], as well as analog VLSI neural systems[49].

Note that our proposed Gaussian synapses naturally require operation in subthreshold regime in order to exploit the exponential feature in the transfer characteristics of the n-type and p-type transistors. Furthermore, as shown in Fig. 1f, Gaussian synapses maintain their characteristic features even when the supply voltage ($V_{DS}$) is scaled down to 200 mV or beyond. This allows the Gaussian synapses to be inherently low power. For the proof-of-concept demonstration of Gaussian synapses, we have used relatively thicker back-gate and top-gate oxides with respective thicknesses of 285 nm and 120 nm. This necessitates the use of rather large back-gate and top-gate voltages in the range of $-50$ V to 50 V and $-35$ V to 35 V, respectively. The power consumption by our proof-of-concept Gaussian synapses will still be high in spite of scaling the supply voltage and exploiting the subthreshold current conduction in the range of nano amperes through the MoS$_2$ and BP FETs. This is because power consumption by Gaussian synapse is simply the area under the $I_D$ versus $V_{BG}$ curve. By scaling the thicknesses of both the top and bottom gate dielectrics, it is possible to scale the operating gate voltages and thereby achieve desirable power benefits from the Gaussian synapses. Ultra-thin dielectric materials such as Al$_2$O$_3$ and HfO$_2$, which offer much larger dielectric constants, $\varepsilon_{ox}$, of $\approx 9$ and 25, respectively, are now routinely used as gate oxides for highly scaled Si FinFETs[50]. It must also be emphasized that the use of atomically thin 2D materials allows geometric miniaturization of Gaussian synapses without any loss of electrostatic integrity, which aids size scaling. We would like to remind the readers that the scalability of FETs is captured through a simple parameter called the screening length ($\lambda_{SC} = \sqrt{\frac{\varepsilon_{body}}{\varepsilon_{ox}} t_{body} t_{ox}}$), which determines the decay of the potential (band bending) at the source/drain contact interface into the semiconducting channel. In this expression, $t_{body}$ and $t_{ox}$ are the thicknesses and $\varepsilon_{body}$ and $\varepsilon_{ox}$ are the dielectric constants of

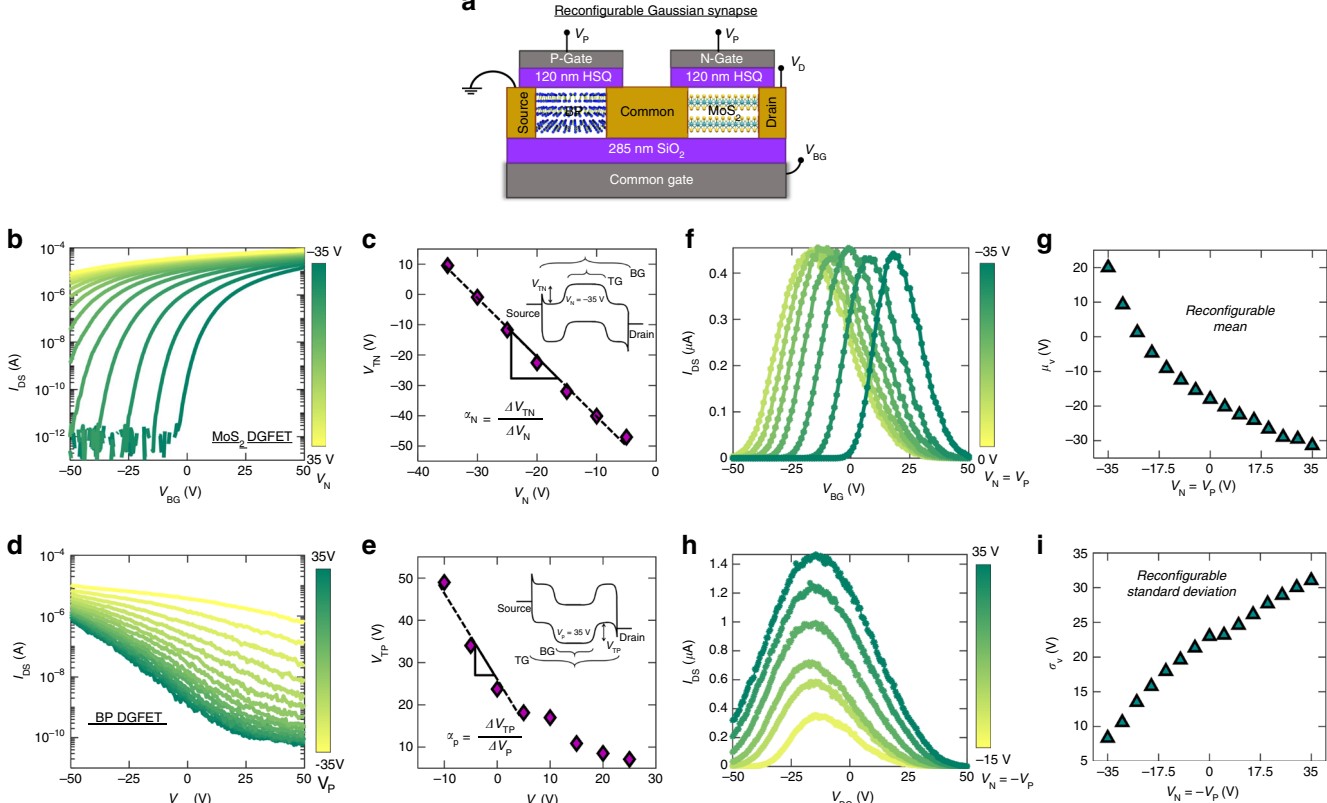

**Fig. 2** Reconfigurable Gaussian synapse. **a** Schematic of a reconfigurable Gaussian synapse involving dual-gated n-type MoS$_2$ FET and p-type BP FET. The top-gate stack was fabricated using hydrogen silsesquioxane (HSQ) as the top-gate dielectric and nickel/gold (Ni/Au) as the top-gate electrode. **b** Back-gated transfer characteristics of the MoS$_2$ FET at $V_D = 1$ V, for different top-gate voltages ($V_N$). **c** Back-gate threshold voltage, $V_{TN}$ of MoS$_2$ FET as function of $V_N$ was extracted using the constant current method. Inset shows the band diagram elucidating how $V_N$ controls $V_{TN}$ by electrostatically adjusting the height of the thermal energy barrier for electron injection into the MoS$_2$ channel. **d** Back-gated transfer characteristics of the BP FET at $V_D = 1$ V, for different top-gate voltages ($V_P$). **e** Back-gate threshold voltage, $V_{TP}$ of BP FET as function of $V_P$. Inset shows the band diagram elucidating how $V_P$ controls $V_{TP}$ by electrostatically adjusting the height of the thermal energy barrier for hole injection into the BP channel. As expected the slope ($\alpha_N = 1.91$) of $V_{TN}$ versus $V_N$ and the slope ($\alpha_P = 2$) of $V_{TP}$ versus $V_P$ are found to be similar and equal to the ratio of top-gate capacitance ($C_{TG}$) to the back-gate capacitance ($C_{BG}$), which in our case was ≈1.94. **f** Transfer characteristics of the Gaussian synapse for different values of $V_N = V_P$. This configuration allows us to shift the mean ($\mu_V$) of the Gaussian synapse without changing the amplitude ($A$) and the standard deviation ($\sigma_V$). **g** $\mu_V$ as a function of $V_N = V_P$. **h** Transfer characteristics of the Gaussian synapse for different values of $V_N = -V_P$. This configuration allows us to configure $\sigma_V$ while keeping $\mu_V$ constant. **i** $\sigma_V$ as a function of $V_N = -V_P$. However, this configuration also results in an increase in the amplitude ($A$) of the Gaussian synapse as $\sigma_V$ increases. This increase can be adjusted by changing the drain voltage ($V_D$) since $A$ is linearly proportional to $V_D$. Nevertheless, by controlling $V_N$, $V_P$ and $V_D$, it is possible to adjust the mean, standard deviation and amplitude of the Gaussian synapse

the semiconducting channel and the insulating oxide, respectively. As discussed by Frank et al.[51], to avoid short channel effects the channel length of an FET ($L_{CH}$) has to be at least three times higher than the screening length ($\lambda_{SC}$). For atomically thin semiconducting monolayers of 2D materials, $t_{body}$ is 0.6 nm, which corresponds to $\lambda_{SC}$ of 1.3 nm, whereas, for the most advanced FinFET technology the thickness of Si fins can be scaled down to only 5 nm without severely increasing the bandgap due to quantum confinement effects and reducing the mobility due to enhanced surface roughness scattering. Nevertheless, the above discussions, clearly articulate how BP/MoS$_2$ 2D heterostructure based Gaussian synapses can facilitate effortless hardware realization of PNNs and thereby aid complexity scaling without compromising energy and size scaling.

**Reconfigurable Gaussian synapse.** For the hardware implementation of Gaussian synapses, it is highly desirable to demonstrate complete tunability of the device transfer function, i.e., $A$, $\mu_V$, and $\sigma_V$ of the Gaussian distribution. This could be

achieved, seamlessly, in our device structure *via* threshold engineering through additional gating of either or both MoS$_2$ and BP FETs. Fig. 2a shows the schematic representation of a reconfigurable Gaussian synapse, where, both MoS$_2$ and BP FETs are dual-gated (DG). The top-gate stack was fabricated using hydrogen silsesquioxane (HSQ)[52,53] as the top-gate dielectric with nickel/gold (Ni/Au) as the top-gate electrode. The fabrication process flow is described in the experimental method section. Fig. 2b shows the experimentally measured back-gated transfer characteristics of the MoS$_2$ FET at $V_D = 1$ V for different top-gate voltages ($V_N$). Clearly, $V_N$ controls the back-gate threshold voltage, $V_{TN}$ of the MoS$_2$ FET as shown in Fig. 2c. The energy band diagram shown in the inset of Fig. 2c can be used to explain the concept of threshold voltage engineering using gate electrostatics. The top-gate voltage determines the height of the potential barrier for electron injection inside the MoS$_2$ channel, which must be overcome by applying a back-gate voltage to enable current conduction from the source to the drain terminal. Negative top-gate voltages increase the potential barrier for electron injection and hence necessitate larger positive back-gate voltages to achieve

similar level of current conduction. As such, $V_{TN}$ becomes more positive (less negative) for large negative $V_N$. Note that the slope ($\alpha_N$) of $V_{TN}$ versus $V_N$ in Fig. 2c must be proportional to the ratio of top-gate capacitance ($C_{TG}$) to the back-gate capacitance ($C_{BG}$). This follows directly from the principle of charge balance, which ensures that the channel charge induced by the top-gate voltage must be compensated by the back-gate voltage at threshold. We extracted the value for $\alpha_N$ to be 1.91. This is consistent with the theoretical prediction of approximately 1.94, given that the top-gate and back-gate dielectric thicknesses are 120 nm and 285 nm, respectively and the top-gate insulator, HSQ, has a slightly lower dielectric constant ($\approx$3.2) than the back-gate insulator, $SiO_2$ (3.9). Fig. 2d shows the experimentally measured back-gated transfer characteristics of the BP FET at $V_D = 1$ V, for different top-gate voltages ($V_P$). As expected, $V_P$ controls the back-gate threshold voltage, $V_{TP}$ of the BP FET as shown in Fig. 2e. Here, the top-gate voltage influences the height of the potential barrier for hole injection, which is overcome by applying a back-gate voltage, enabling current conduction from the drain to the source terminal. The corresponding energy band diagram is shown in the inset of Fig. 2e. Positive top-gate voltages increase the potential barrier for hole injection and hence necessitate larger negative back-gate voltages to achieve similar level of current conduction. As such, $V_{TP}$ becomes more negative (less positive) for large positive $V_P$ values. We also extracted the slope ($\alpha_P$) of $V_{TP}$ versus $V_P$ in Fig. 2e and, as expected, found a similar value of $\approx$2.

The dual-gated $MoS_2$ and BP FETs allow complete control of the shape of the Gaussian synapse. Fig. 2f shows the experimentally measured transfer characteristics of the Gaussian synapse for different values of $V_N = V_P$. This configuration allows us to shift the mean ($\mu_V$) of the Gaussian synapse without changing the amplitude ($A$) or the standard deviation ($\sigma_V$) Fig. 2g shows $\mu_V$ plotted as a function of $V_N = V_P$. We are able to do this since the back-gate threshold voltages for both $MoS_2$ and BP FETs shift in the same direction in this configuration. Similarly, Fig. 2h shows the experimentally measured transfer characteristics of the Gaussian synapse for different values of $V_N = -V_P$. Under this configuration, the back-gate threshold voltages for $MoS_2$ and BP FETs shift in opposite directions. As such the $\mu_V$ of the Gaussian distribution remains constant, whereas, $\sigma_V$ keeps increasing. Fig. 2i shows $\sigma_V$ plotted as a function of $V_N = -V_P$. However, this configuration also results in an increase in the amplitude ($A$) of the Gaussian synapse as $\sigma_V$ increases. This increase can be adjusted by changing the drain voltage ($V_D$) since $A$ is linearly proportional to $V_D$. Nevertheless, by controlling $V_N, V_P$, and $V_D$, it is possible to adjust the mean, standard deviation, and amplitude of the Gaussian synapse.

**Scaled Gaussian synapses**. In order to project the performance of scaled Gaussian synapses, we used the Virtual Source (VS) model that was originally developed by Khakifirooz, A. et al. for short channel Si MOSFETs[54]. for short channel Si MOSFETs. The Gaussian transfer characteristics ($I_D$ versus $V_G$ for different $V_D$) were simulated in the following Eqs. 4, 5, and 6. In the VS model, both the subthreshold and the above threshold behavior is captured through a single semi-empirical and phenomenological relationship that describes the transition in channel charge density from weak to strong inversion (Eq. 5).

$$I_D = \frac{V_D}{R_N + R_P}; \quad R_N = \frac{L_N}{W_N}\frac{1}{\mu_N Q_N}; \quad R_P = \frac{L_P}{W_P}\frac{1}{\mu_P Q_P}; \quad (4)$$

$$Q_N = C_{BG} m \frac{k_B T}{q} \log\left[1 + \exp\left(\frac{V_G - V_{TN}}{mk_B T/q}\right)\right];$$
$$Q_P = C_{BG} m \frac{k_B T}{q} \log\left[1 + \exp\left(-\frac{V_G - V_{TP}}{mk_B T/q}\right)\right]; \quad (5)$$

$$V_{TN} = \alpha V_N; \quad V_{TP} = \alpha V_P; \quad \alpha = C_{TG}/C_{BG}; \quad (6)$$

Here, $R_N$ and $R_P$ are the resistances, $L_N$ and $L_P$ are the lengths, $W_N$ and $W_P$ are the widths, $\mu_N$ and $\mu_P$ are the carrier mobility values, and $Q_N$ and $Q_P$ are the inversion charges corresponding to the n-type and the p-type 2D-FETs, respectively. The band movement factor $m$ can assumed to be unity for a fully depleted and ultra-thin body 2D-FET with negligible interface trap capacitance. Finally, $V_{TN}$ and $V_{TP}$ are the threshold voltages of the n-type and p-type 2D FETs determined by their respective top-gate voltages $V_N$ and $V_P$. Note, that in the subthreshold regime, the inversion charges i.e., $Q_N$ and $Q_P$ increase exponentially with $V_G$, whereas above threshold, the inversion charge is a linear function of $V_G$, which is seamlessly captured through the VS model. Fig. 3a, b show the simulated transfer characteristics of the individual n-type and p-type 2D FETs, respectively, and Fig. 3c shows the transfer characteristics of the Gaussian synapse based on their heterostructure for different combinations of the top-gate voltages, following the VS model, as described above. Furthermore, Fig. 3d–f, respectively, demonstrate the tunability of $A, \mu_V$, and $\sigma_V$ of the Gaussian synapse via top-gate voltages $V_N$ and $V_P$. More details on the design of Gaussian synapses can be found in the supplementary information section.

**Brainwave classification**. Next, we show simulation results suggesting that PNNs based on Gaussian synapses can be used for the classification of various neural oscillations, also known as the brainwaves that are fundamental to human awareness, cognition, emotions, and actions. These rhythmic and repetitive oscillations that originate from synchronous and complex firing of neural ensembles are observed throughout the central nervous system and are essential in controlling the neuro-physiological health of any individual. As shown in Fig. 4a, the brainwaves are divided into five frequency bands based on the present understanding and interpretation of their functions. Lower frequency (0.5–3.5 Hz) and higher amplitude delta waves ($\delta$) are generated during deepest meditation and dreamless sleep, suspending all external awareness and facilitating healing and regeneration. Disruptions in $\delta$-wave activity can lead to neurological disorders such as dementia, schizophrenia, parasomnia, epilepsy, and Parkinson's disease. Low frequency (4–8 Hz) and high amplitude theta waves ($\theta$) originate during sleep or deep meditation with senses withdrawn from the external world and focused within. Normal firing of $\theta$-waves enables learning, memory, intuition, and introspection, while excessive activity can lead to attention-deficit/hyperactivity disorder (ADHD). Mid-frequency (8–14 Hz) and mid-amplitude alpha waves ($\alpha$) represent the resting state of the brain and facilitate mind/body coordination, mental peace, alertness, and learning. High frequency (16–32 Hz) and low amplitude beta waves ($\beta$) dominate the wakeful state of consciousness, direct our concentration towards cognitive tasks such as problem solving and decision-making, and at the same time consume a tremendous amount of energy. In clinical context, $\beta$-waves can be used as biomarkers as they indicate the release of gamma aminobutyric acid, the principal inhibitory neurotransmitter in the mammalian nervous system. Finally, gamma waves ($\gamma$) are the fastest (32–64 Hz) and quietest brain waves. These waves were dismissed as neural

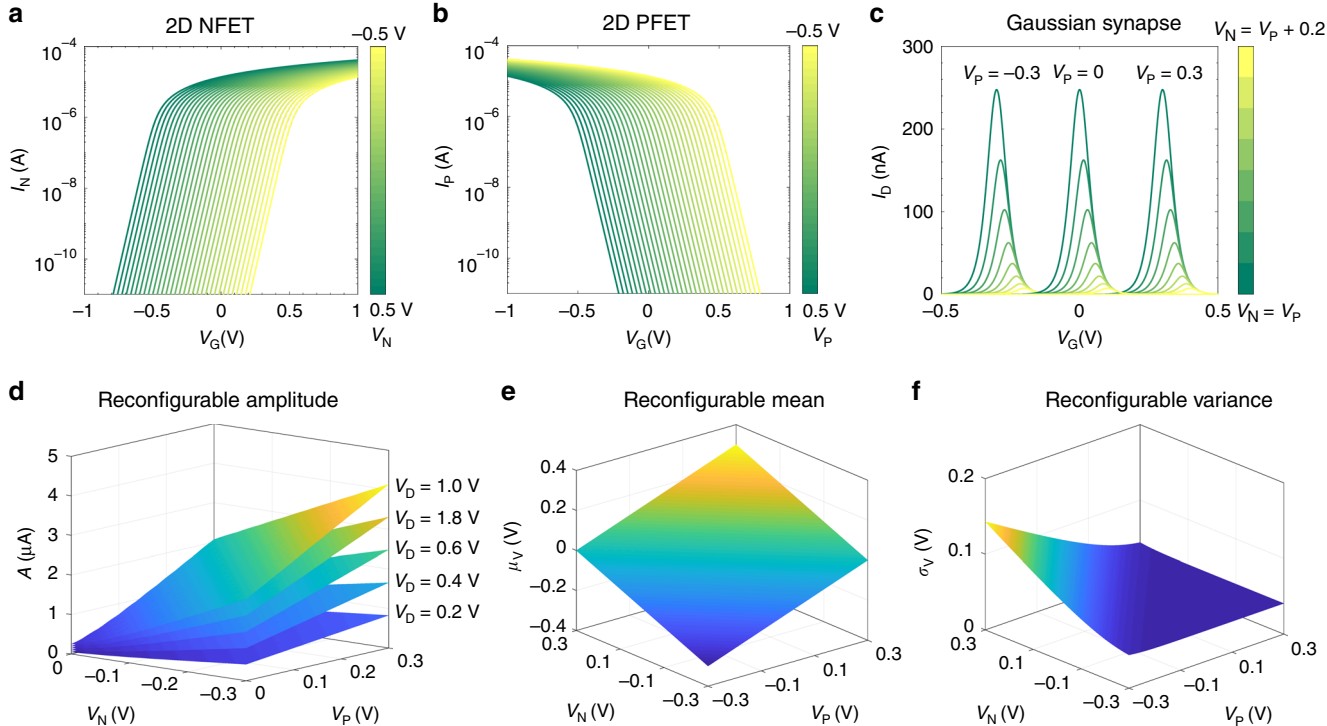

**Fig. 3** Scaled Gaussian Synapses. Simulated back-gated transfer characteristics of (**a**) n-type and (**b**) p-type 2D FET for corresponding different top-gate voltages $V_N$ and $V_P$, respectively, using the Virtual Source (VS) model. In the VS model, both the subthreshold and the above threshold behavior are captured through a single semi-empirical and phenomenological relationship that describes the transition in channel charge density from weak to strong inversion. **c** Transfer characteristics, (**d**) amplitude, (**e**) mean, and (**f**) standard deviation of the Gaussian synapse obtained via heterogeneous integration of the n-type and the p-type 2D FETs for different combinations of $V_N$ and $V_P$. The following parameters were used for the simulation. $L_N = L_P = 1\,\mu m$; $W_N = W_P = 2\,\mu m$; $\mu_N = \mu_P = 20\ cm^2(Vs)^{-1}$; $C_{BG} = 7\times10^{-3}\,F\,m^{-2}$; $M = 1$; $\alpha = 1$; $V_D = 1\,V$

noise until recently, when researchers discovered the connection to greater consciousness and spiritual activity culminating in the state of universal love and altruism[55]. The above discussion clearly shows the immense importance of brainwaves in regulating our daily experience. Instabilities in brain rhythm can be catastrophic, leading to insomnia, narcolepsy, panic attacks, obsessive-compulsive disorder, agitated depression, hyper-vigilance, and impulsive behaviors. Early diagnosis of abnormal brainwave activity through neural networks can help prevent chronic neuro diseases and mental and emotional disorders.

Figure 4b, c, and 4d show the frequency pattern of the normalized power spectral density (PSD) for each type of brainwaves, extracted from the Fast Fourier Transform (FFT) of the time domain Electroencephalography (EEG) data (sequential montage) with increasing sampling times that correspond to sample sizes of $N = 512$, 2560, and 6400, respectively. Clearly, as the training set becomes more and more exhaustive, the discrete frequency responses corresponding to each type of brainwave evolve into continuous spectrums that show complex patterns. Furthermore, the system is highly nonlinear with functional dependence of the PSDs on frequency being rather complicated for each type of brainwave. As such, classification of brainwave patterns using conventional ANNs, can be challenging[56–58]. In addition, ANNs require optimum training algorithms and extensive feature extraction and preprocessing of the training sample in order to achieve reasonable accuracy. In contrast, as demonstrated in Fig. 4d, the PNN adopts single pass learning by defining the class PDF for each of the brainwave pattern in the frequency domain using Gaussian mixture model (GMM)[59,60]. GMM is represented as the weighted sum of a finite number of scaled (different variance) and shifted (different mean) normal

distributions as described by Eq. 2.

$$p(x) = \sum_{i=1}^{K} \psi_i N[x/\mu_i, \sigma_i]; N[x/\mu_i, \sigma_i]$$
$$= \frac{1}{\sqrt{2\pi\sigma_i^2}} \exp\left[-\frac{(x-\mu_i)^2}{2\sigma_i^2}\right]; \sum_{i=1}^{K} \psi_i = 1 \quad (7)$$

A GMM with $K$ components is parameterized by two types of values, the component weights ($\psi_i$) and the component means ($\mu_i$) and variances ($\sigma_i^2$) with the constraint that $\sum_{i=1}^{K} \psi_i = 1$, so that the total probability distribution normalizes to unity. For each type of brainwave pattern, the GMM parameters for the $K$ component were estimated from the training data corresponding to $N = 25,600$, using the non-linear least square method. Figure 4e shows root mean square errors (RMSEs) calculated as a function of $K$, i.e., the number of Gaussian curves used in the corresponding GMMs. Clearly, a very limited number of Gaussian functions are necessary to capture the non-linear decision boundary for each of the brainwaves. This enormously reduces the energy and size constraint for the PNNs based on Gaussian synapses.

Finally, Fig. 5a shows simulation results evaluating the PNN architecture for the detection of new brainwave patterns. The PNN consists of 4 layers: input, pattern, summation, and output. The amplitude of the new FFT data is relayed from the input layer to the pattern layer as the drain voltage ($V_D$) of the Gaussian synapses, whereas, the frequency range (0–64 Hz) is mapped to the back-gate voltage ($V_G$) range. The summation layer integrates the current over the full swing of $V_G$ from the individual pattern blocks and communicates with the winner-takes-it-all (WTA)

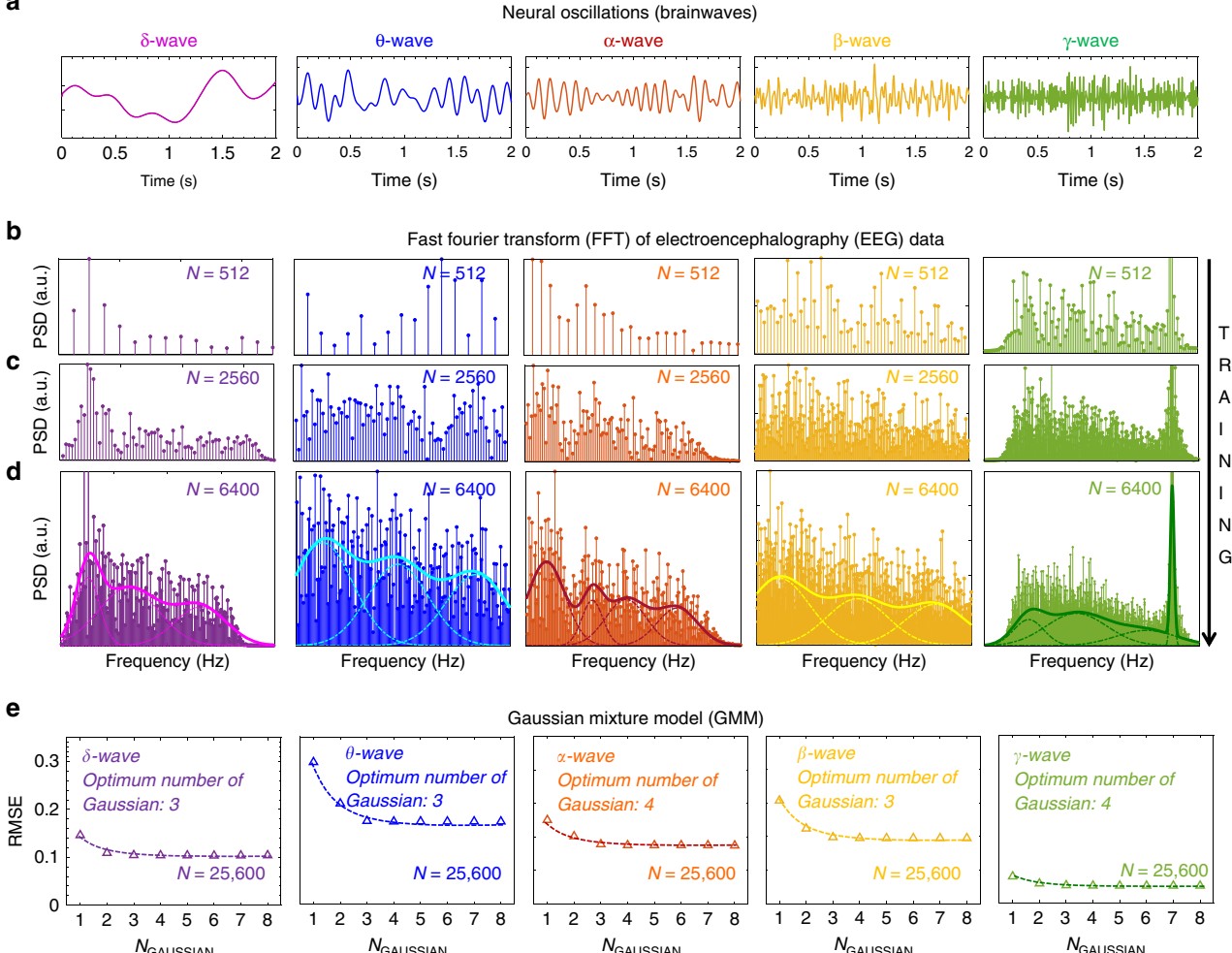

**Fig. 4** Recognition of Brainwaves using Gaussian Mixture Model (GMM). **a** Brainwaves are divided into five frequency bands. 0.5–3.5 Hz: Delta waves (δ), 4–8 Hz: Theta waves (θ), 8–14 Hz: Alpha waves (α), 16–32 Hz: Beta waves (β) and 32–64 Hz: Gamma waves (γ). Normalized power spectral density (PSD) as a function of frequency for each type of brainwaves, extracted using the Fast Fourier Transform (FFT) of the time domain Electroencephalography (EEG) data (sequential montage) with increasing sampling times that correspond to sample sizes of (**b**) $N = 512$, (**c**) $N = 2560$, and (**d**) $N = 6400$. As the training set becomes more and more exhaustive, the discrete frequency responses evolve into continuous spectrums. The highly nonlinear functional dependence of the PSDs on frequency for each type of brainwave can be classified using GMM, which is represented as the weighted sum of a finite number of scaled (different variance) and shifted (different mean) normal distributions. **e** Root mean square error (RMSE) as a function of K, i.e., the number of Gaussian used in the GMMs for each type of brainwave. Clearly, a very limited number of Gaussian synapses are necessary to capture the non-linear decision boundaries

circuit that allows the output layer to recognize the brainwave patterns. We implemented our PNN architecture on 10 whole-night polysomnographic recordings, obtained from 10 healthy subjects in a sleep laboratory using a digital 32-channel polygraph (details can be found in the method section). The percentage of different brainwave components as recognized by the PNN are shown as a color map in Fig. 5b. As expected, the PNN recognizes the dominant presence of delta and theta waves in the sleep samples. Furthermore, Fig. 5c shows the total power consumption (details can be found in the supplementary information section) by the PNN as a function of the supply voltage ($V_{DD}$) and sample volume (N). As expected, the power dissipation scales with N and $V_{DD}$. Interestingly, even for a large sample volume of $N = 2 \times 10^5$, corresponding to 8 h of EEG data, the power consumption by the proposed PNN architecture was found to be as frugal as 3 μW for $V_{DD} = 0.1$ V, which increases to only 350 μW for $V_{DD} = 1.0$ V. A direct comparison of power dissipation with digital CMOS will be premature at this time, especially since the peripheral circuits required for the proposed PNN architecture

will add power dissipation overhead. Nevertheless, these preliminary results show that the PNN architectures based on Gaussian synapses can offer extreme energy efficiency.

## Discussion

In conclusion, we have demonstrated reconfigurable Gaussian synapses based on the heterostructure of atomically thin 2D layered semiconductors as a new class of analog and probabilistic computational primitives that can reinstate both energy and size scaling aspects of computation. Furthermore, we elucidated how Gaussian synapses enable direct hardware realization of PNNs, which offer simple and effective solutions to a wide range of pattern classification problems and thereby resurrect complexity scaling. Finally, we show simulation results suggesting that PNN architecture based on Gaussian synapses is capable of recognizing complex neural oscillations or brainwave patterns from large volumes of EEG data with extreme energy efficiency. We believe that these findings will foster the much-needed interest in

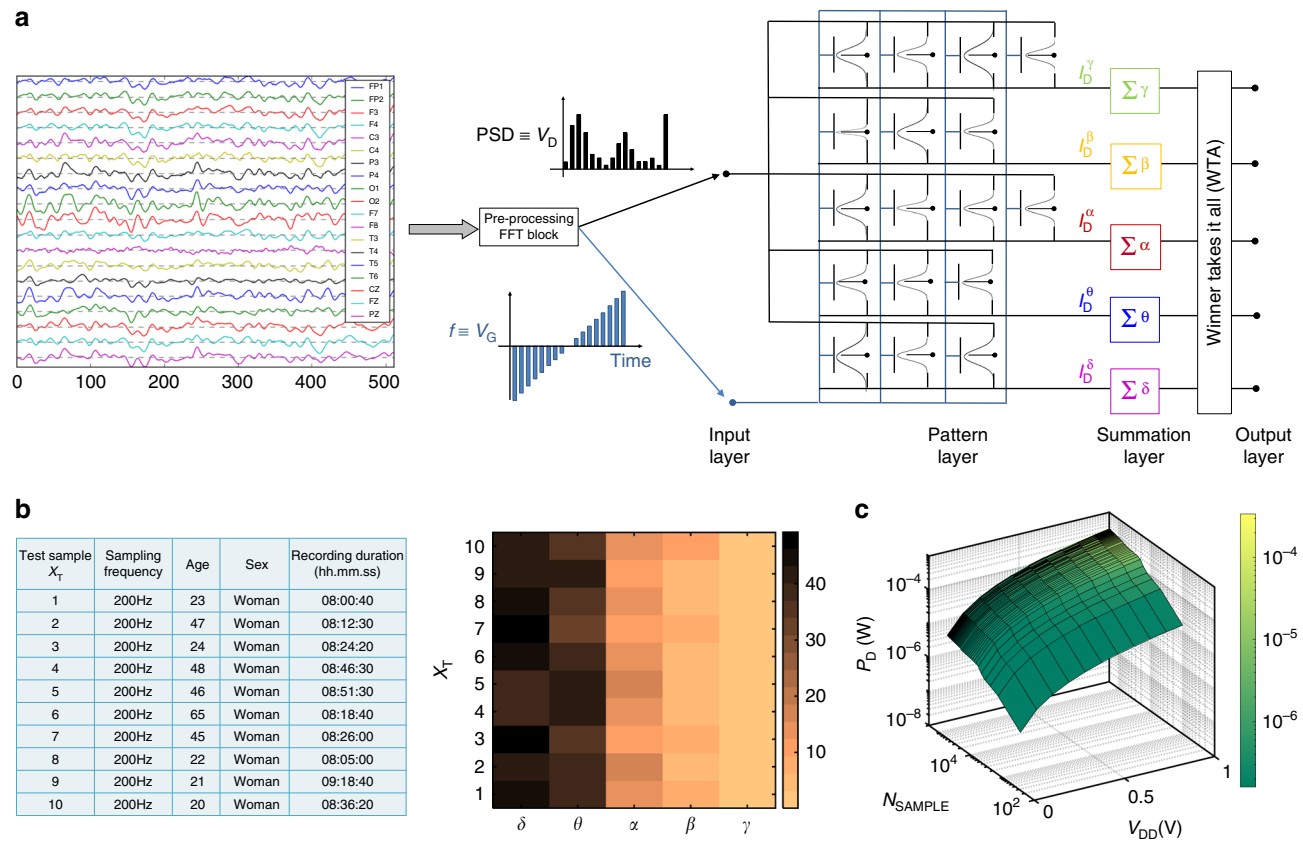

**Fig. 5** PNN Architecture for Brainwave Recognition. **a** The PNN consists of 4 layers: input, pattern, summation, and output. The amplitude of the FFT data is relayed from the input layer to the pattern layer as drain voltage ($V_D$) of the Gaussian synapses, whereas, the frequency range (0–64 Hz) is mapped to the back-gate voltage ($V_G$) range. The summation layer integrates the current over the full swing of $V_G$ from the individual pattern blocks and communicates with the winner-takes-it-all (WTA) circuit that allows the output layer to recognize the brainwave patterns. **b** Implementation of PNN Architecture: 10 whole-night polysomnographic recordings and the corresponding outcome of the PNN architecture shown using a color map. The PNN recognizes the dominant presence of delta and theta waves in all the sleep samples. **c** Power Consumption by PNN Architecture: The total power consumption by the PNN as a function of the supply voltage ($V_{DD}$) and sample volume ($N_{SAMPLE}$). As expected, the power dissipation scales with $N_{SAMPLE}$ and $V_{DD}$

hardware implementation of PNNs and ultimately aid high performance and low power computing infrastructure.

## Methods

**Device fabrication and measurements**. $MoS_2$ and BP flakes were micromechanically exfoliated on 285 nm thermally grown $SiO_2$ substrates with highly doped Si as the back-gate electrode. The thicknesses of the $MoS_2$ and BP flakes were in the range of 3–20 nm. $MoS_2$ is a 2D layered material with the lattice parameters $a = 3.15$ A°, $b = 3.15$ A°, $c = 12.3$ A°, $\alpha = 90°$, $\beta = 90°$, and $\gamma = 120°$. The layered nature due to van der Waals (vdW) bonding results in a higher value for c. This enables mechanical exfoliation of the material to obtain ultra-thin layers of $MoS_2$. BP exhibits a puckered honeycomb lattice structure. It has phosphorous atoms existing on two parallel planes. The lattice parameters are given by $a = 3.31$ A°, $b = 10.47$ A°, $c = 4.37$ A°, $\alpha = 90°$, $\beta = 90°$, and $\gamma = 90°$. The Source/Drain contacts were defined using electron-beam lithography (Vistec EBPG5200). Ni (40 nm) followed by Au (30 nm) was deposited using electron-beam (e-beam) evaporation for the contacts. Both devices were fabricated with a channel length of 1 μm. The width of the $MoS_2$ and BP devices were 0.78 μm and 2 μm, respectively. The top-gated devices were fabricated with hydrogen silsesquioxane (HSQ) as the top-gate dielectric. The top-gate dielectric was deposited by spin coating 6% HSQ in methyl isobutyl ketone (MIBK) (Dow Corning XR-1541–006) at 4000 rpm for 45 s and baked at 80 °C for 4 min. The HSQ was patterned using an e-beam dose of 2000 μC cm$^{-2}$ and developed at room temperature using 25% tetramethylammonium hydroxide (TMAH) for 30 s following a 90 s rinse in deionized water (DI). Next, the HSQ was cured in air at 180 °C and then 250 °C for 2 min and 3 min, respectively. The thickness of the HSQ layer, used as the top-gate dielectric was 120 nm. Top-gate electrodes with Ni (40 nm) followed by Au (30 nm) were patterned with the same procedure as the source and drain contacts. Given the instability of BP, we took special care to ensure minimal exposure time to the air

while fabricating BP devices by storing the material in vacuum chambers between different fabrication steps. In addition, all the three lithography steps involved in the device fabrication were done within a period of 3 days. The electrical characterizations were obtained at room temperature in high vacuum (≈10$^{-6}$ Torr) a Lake Shore CRX-VF probe station and using a Keysight B1500A parameter analyzer.

**EEG data**. The data used for our study were obtained from the DREAMS project, which were acquired in a sleep laboratory of a Belgium hospital using a digital 32-channel polygraph (BrainnetTM System of MEDATEC, Brussels, Belgium). They consist of whole-night polysomnographic recordings, coming from healthy subjects. At least two EOG channels (P8—A1, P18—A1), three EEG channels (CZ—A1 or C3—A1, FP1—A1, and O1—A1), and one submental EMG channel were recorded. The standard European Data Format (EDF) was used for storing. The sampling frequency was 200 Hz. These recordings have been specifically selected for their clarity (i.e., that they contain few artifacts) and come from persons, free of any medication, volunteers in other research projects, conducted in the sleep lab.

## Data availability
The data that support the plots within this paper and other findings of this study are available from the corresponding authors upon reasonable request.

## Code availability
The codes used for data analysis are available from the corresponding authors upon request.

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

## Acknowledgements

The authors would like to acknowledge the contribution of Joseph R Nasr, Harikrishnan Ravichandran, Harikrishnan Jayachandrakurup, and Sarbashis Das for their help in device fabrication. This work was partially supported through Grant Number FA9550–17–1–0018 from Air Force Office of Scientific Research (AFOSR) through the Young Investigator Program.

## Author contributions

S.D. conceived the idea, designed the experiments and wrote the paper. A.S., S.S.R., and A.P. performed the experiments. All authors analyzed the data, discussed the results, agreed on their implications, and contributed to the preparation of the paper.

## Additional information

**Competing interests:** The authors declare no competing interests.

**Peer Review Information** Nature Communications thanks Qing Wan, Su-Ting Han and other, anonymous, reviewers for their contribution to the peer review of this work. Peer reviewer reports are available.

