## [Peer Review File · Nature Communications]

Reviewers' comments:

Reviewer #1 (Remarks to the Author):

Comments to the Author

In this manuscript (entitled "Gaussian Synapses for Probabilistic Neural Network"), the authors proposed a reconfigurable Gaussian synapse based on heterostructures of atomically thin two dimensional (2D) layered materials for probabilistic computational primitive. Further, how such a Gaussian synapse can defeat the Boltzmann limit to reinstate energy scaling, quantum limit to restore size scaling and von Neumann limit to facilitate complexity scaling was discussed. At last, the classification of the brainwave patterns was demonstrated by the Gaussian synapse based probabilistic neural network. The results reported here are very interesting for building probabilistic neural network. This manuscript needs to be greatly improved before it can be accepted. There are some comments to the authors.

(1) In Figure 1(g) and Figure 1(h), the off state current of the devices is very large (~10⁻⁶A)? What causes this current? Can it be reduced?

(2) MoS₂ and BP flakes were micromechanically exfoliated in this manuscript. What is the biggest size of the flakes? How to integrate a complex system on such micromechanically exfoliated flakes?

(3) In this manuscript, a probabilistic neural network was built for brainwave recognition. Is there any commercial product for brainwave recognition which can help prevent chronic neuro diseases and mental and emotional disorders? If there is, please compare it with the proposed probabilistic neural network.

(4) In Figure 2(d), (e), and (f), V_D is 0.2V, 0.6V, and 1.0V, respectively. However, in the text, "Figure 2d, 2e, and 2f, respectively, show the experimentally measured transfer characteristics of the Gaussian synapse corresponding to V_D = 0.2V, 0.6V and 0.8V for different applied V_N." The value of V_D is inconsistent.

Reviewer #2 (Remarks to the Author):

The authors report a new type of analog devices named as Gaussian synapses, based on the heterostructure of atomically thin 2D layered semiconductors, which enable the direct hardware realization of probabilistic neural networks (PNNs). They also demonstrate that PNN architecture based on Gaussian synapses is capable of recognizing complex neural oscillations or brainwave patterns from large volumes of Electroencephalography (EEG) data with extreme energy efficiency. The work is interesting and the experimental demonstration of Gaussian synapses with the heterogeneous integration of n-type MoS₂ and p-type black phosphorus (BP) based transistors is ingenious. The reviewer suggests major revisions before acceptance of the manuscript, with the following comments to be addressed:

1. The experimental demonstration of Gaussian synapses based on atomically thin 2D layered semiconductors is the cornerstone of this work. However, this manuscript lacks quantitative experimental characterization of both materials and devices. Such as thickness, size, and lattice parameters of 2D materials, the channel W/L of the devices, et al.
2. Black phosphorus is expected to oxidize upon exposure to air. The formation of surface oxide species may cause the increase in surface roughness and nonuniform degradation, thus reduced field-effect transistor performance. How to avoid the structural degradation and its effect on the electronic properties when fabricate the top-gated devices?
3. The authors mentioned that their choice of MoS₂ and BP was due to their highest n-type and p-type performing. However, the carrier mobility of the n-type and p-type transistor were not balanced. Could the authors comment how this affects the reconfigurability of the Gaussian synapse? Could they design further experiments for the device optimization?
4. By top gating the MoS₂ FET, the device transfer function can be tuned. While the

reconfigurability of the Gaussian synapses can be further extended if the BP FET is also top-gated. The authors should provide the transfer characteristics of the BP DGFET.

5. The authors should explain clearly the working principle of the Gaussian synapse and the role of the Common terminal in the Figure 1e and Figure 2a.

6. Fig 2 (b) shown the transfer characteristics of MoS2 DGFET for different top gate voltages (VN), the authors should explain this phenomenon from the energy level, interface analysis and add more discussions about the mechanisms.

7. The authors claims the slope (α) of VT versus VN in figure 2c is proportional to the ratio of top-gate capacitance (CTG) to the back-gate capacitance (CBG), which was found to be $\sim 2/3$. It lacks adequate evidence. VT versus VN is related to the interface between MoS2 and HSQ or the energy level of MoS2, when the interface is different, the charge trapped capability is different and the ratio of VT and VN will be different. So, the authors need to further study the interface between MoS2 and HSQ. In other words, only when the interface of MoS2 and HSQ is the same as the interface of MoS2 and SiO2, the slope (α) of VT versus VN may compare with the ratio of top-gate capacitance (CTG) to the back-gate capacitance (CBG).

8. The authors claimed that the PNN architectures based on Gaussian synapses can offer extreme energy efficiency. They use a Virtual Source (VS) model to project the performance of scaled Gaussian synapse. However, it is not guaranteed that a scaled device will show similar behaviors with a lowered energy consumption. Also, the authors need to comment on the energy consumption and compare them to literature to highlight the advantage of this work.

9. The experimental results were all obtained from single devices. However, the realization of PNN in more complicated programs require more Gaussian synapses. The device variation is a critical issue for the achievement of a large-size array. Was the reliability and uniformity issue of the devices?

10. On the Brainwave Recognition : the authors state that the PNN recognizes the dominant presence of delta and theta waves in the sleep samples, almost with 100% accuracy. How about the accuracy when conceive of much more complex sample selection?

11. There are many format errors in the text and references. For example, ref 1/ref 9, ref 8/ref 63 is the same. There are some typos and grammatical mistakes in the paper, for example, "mersister" on the page 4 line 6.

Reviewer #3 (Remarks to the Author):

This paper describes a new electronic synapse created as a combination of n-type and p-type back-gate field-effect transistors in a single integrated device, including means to tune the resulting Gaussian-like characteristic. Experimental measurements of the device are included, and a model is then simulated to suggest classification of brain-wave data.

The contribution in device development may be significant but the application to brain-wave classification is ad hoc, and its demonstration far-fetched. The Fourier transforms of EEG synthesized data are computed and then the simulated spectra are presented as voltage to a collection of simulated synapses to implement a gaussian mixture model.

In all, this paper makes bold claims on a demonstration of neuromorphic computing beating Boltzmann and von Neumann limits and achieving brain-wave detection that are not substantiated. A much more concrete and focused demonstration of the device electronic capabilities would serve the reader much better than the random jumble of deep learning and thermodynamic jargon without substance.

Response to Reviewers' Comments

Reviewer's Comment

Our Response

Changes Made in the Manuscript

Reviewer #1 (Remarks to the Author)

Comments to the Author

In this manuscript (entitled “Gaussian Synapses for Probabilistic Neural Network”), the authors proposed an reconfigurable Gaussian synapse based on heterostructures of atomically thin two dimensional (2D) layered materials for probabilistic computational primitive. Further, how such a Gaussian synapse can defeat the Boltzmann limit to reinstate energy scaling, quantum limit to restore size scaling and von Neumann limit to facilitate complexity scaling was discussed. At last, the classification of the brainwave patterns was demonstrated by the Gaussian synapse based probabilistic neural network. The results reported here are very interesting for building probabilistic neural network. This manuscript needs to be greatly improved before it can be accepted. There are some comments to the authors.

We are glad that the reviewer finds this work very interesting. We would like to thank the reviewer for taking the time to read the manuscript and provide insightful feedback that has helped us greatly in improving the quality of the manuscript. We have provided point-by-point answers to the comments and concerns raised by the reviewer below. We have also highlighted the corresponding changes made to the manuscript.

- 1. In Fig. 1(g) and Fig. 1(h), the off state current of the devices is very large ($\sim 10^{-6}$ A)? What causes this current? Can it be reduced?**

We believe that there is a misunderstanding. Note that in Fig. 1(g) and 1(h), the drain current (I_D) is plotted using the units of μA (10^{-6} A). Hence, the actual off-current (I_{OFF}) of the device is on the order of $10^{-6} \mu\text{A}$ or 10^{-12} A. The I_{OFF} current is rather low and is primarily determined by the noise

floor of the measurement instrument, which in this case is a Keysight B1500 semiconductor parameter analyzer.

In order to avoid confusion, we have now changed the unit of current to Amperes (A) in all corresponding figures in the revised manuscript.

2. MoS₂ and BP flakes were micromechanically exfoliated in this manuscript. What is the biggest size of the flakes? How to integrate a complex system on such micromechanically exfoliated flakes?

The typical size of the flakes obtained from micromechanical exfoliation is around 2-10 μm . We agree with the reviewer that micromechanical exfoliation is not a scalable manufacturing process for large-scale integrated circuits, which involve millions of transistors. Hardware implementation of complex systems such as the probabilistic neural networks (PNNs) using Gaussian synapses will necessitate large-area growth of MoS₂ and BP. Fortunately, recent years have seen tremendous progress in wafer-scale growth of high quality MoS₂ and BP using chemical vapor deposition (CVD) and metal organic chemical vapor deposition (MOCVD) techniques [1-5]. Furthermore, while we have used MoS₂ and BP for our proof-of-concept demonstration of Gaussian synapses, there are 2D materials, such as, WSe₂, which offer ambipolar transport, i.e. the presence of both electron and hole conduction [6] and can be grown over large area using CVD techniques [7]. However, the performance of WSe₂ based n-type and p-type FETs are limited by the presence of large Schottky barriers at the metal/2D contact interfaces [8]. By resolving the contact resistance related issues and improving the quality of large-area synthesized WSe₂, it is possible to implement Gaussian synapses based on WSe₂ as well. Moreover, in recent years several groups have demonstrated p-type MoS₂ and n-type BP, by implementing smart contact engineering and/or doping strategies [9, 10]. Therefore, very-large-scale integration of Gaussian synapses based on CVD grown single 2D material will be possible in the near future for the hardware realization of PNNs. Since the focus of this article is to introduce the novel Gaussian synapse and its benefit as a statistical computing primitive, material optimization is beyond the scope and will be investigated in our future studies.

Following discussion was added in the revised manuscript

While our proof-of-concept demonstration of Gaussian synapses is based on exfoliated MoS₂ and BP flakes, it is well known that the micromechanical exfoliation is not a scalable manufacturing process for large-scale integrated circuits. Therefore, hardware implementation of PNNs using Gaussian synapses will necessitate large-area growth of MoS₂ and BP. Fortunately, recent years have seen tremendous progress in wafer-scale growth of high quality MoS₂ and BP using chemical vapor deposition (CVD) and metal organic chemical vapor deposition (MOCVD) techniques [1-5]. Furthermore, while we have used two different 2D materials, MoS₂ and BP, for our demonstration of Gaussian synapses owing to their superior performance as n-type and p-type FETs, respectively, there are 2D materials, such as, WSe₂, which offer ambipolar transport, i.e. the presence of both electron and hole conduction [6] and can be grown over large area using CVD techniques. However, the performance of WSe₂ based n-type and p-type FETs are limited by the presence of large Schottky barriers at the metal/2D contact interfaces. By resolving the contact resistance related issues [8] and improving the quality of large-area synthesized WSe₂, it is possible to implement Gaussian synapses based on WSe₂ as well. Moreover, in recent years several groups have demonstrated p-type MoS₂ and n-type BP, by implementing smart contact engineering and/or doping strategies [9, 10]. Therefore, very-large-scale integration of Gaussian synapses based on CVD grown single 2D material will be possible in the near future for the hardware realization of PNNs. Since the focus of this article is to introduce the novel Gaussian synapse and its benefit as a statistical computing primitive, material optimization is beyond the scope and will be investigated in our future studies.

- 3. In this manuscript, a probabilistic neural network was built for brainwave recognition. Is there any commercial product for brainwave recognition, which can help prevent chronic neuro diseases and mental and emotional disorders? If there is, please compare it with the proposed probabilistic neural network.**

This is an excellent question raised by the reviewer. There are commercial neurofeedback platforms, which analyze EEG data using in-build software programs and force the user to produce certain brainwaves which helps improve their state of mind. For example, Brainworks is a neuro-therapy tool designed to train the brain for personal development and to treat stress, anxiety,

trauma, post-traumatic stress disorder (PTSD) and other medical conditions (<https://brainworksneurotherapy.com/>). There are other commercial EEG analysis software such as EEGLAB by MATLAB, Advanced Source Analysis (<https://www.nitrc.org/projects/asa/>), Besa Research (<http://www.besa.de/>), etc. Open-source software like Brainstorm, MNE and OPENMEEG are also available. While these tools are extremely powerful when it comes to the analysis of EEG signals, they also demand high processing capabilities. Therefore, it is important to develop hardware components that can reduce the computational effort required to analyze the EEG signals. Unfortunately, a very limited amount of research has been dedicated towards such development. Recently, Wang, Y. *et al.*, used field-programmable gate array (FPGA) for on-chip training and classification of epilepsy data using non-linear support vector machine (SVM) model that yielded 94.2% accuracy [11]. Similarly, Saidi, A. *et al.*, used ant colony optimization (ACO) algorithm for classification of epilepsy data with an accuracy of 98.9% [12]. Saleheen, M. *et al.*, developed a real-time seizure detector using multi-layer perceptron (MLP) based artificial neural networks (ANN) with an accuracy of 99.18% [13]. Finally, Atlaf, M. A. B. *et al.* demonstrated a rapid-eye movement (REM) versus non-REM sleep classifier for Alzheimer patients with a sensitivity of 89.8% and specificity of 93.6% using a linear SVM classifier on FPGA platform [14].

With our PNN architecture, we are able to calculate the percentage of different rhythmic activity such as δ , θ , α , β and γ waves, which can determine the physical state of a person. As shown in Fig. R1 the PNN identifies higher percentage of low-frequency δ waves in the EEG samples corresponding to the sleep data, whereas, the awake state data has higher percentage of θ and α waves. The PNN architecture can certainly be extended to create a classifier, which could use the percentage strength of different EEG oscillations to predict the physical state of a person. Since the function of our PNN architecture is to find the percentage strength of different brainwaves, it would not be ideal to compare it to the classifiers

Figure R1. Classification of EEG data using PNN architecture for 5 awake subjects and 5 sleep subjects. PNN rightly identifies higher percentage of low-frequency δ waves in the EEG samples corresponding to the sleep data; whereas, the awake state data has higher percentage of θ and α waves.

described above, which are also used in different contexts. Nevertheless, even for a large sample volume of $N = 2 \times 10^5$, corresponding to 8 hrs of EEG data, the power consumption by our PNN architecture was found to be as frugal as $3 \mu\text{W}$ for $V_{DD} = 0.1\text{V}$, which increases to only $350 \mu\text{W}$ for $V_{DD} = 1.0\text{V}$.

4. In Fig. 2(d), (e), and (f), V_D is 0.2V, 0.6V, and 1.0V, respectively. However, in the text, “Fig. 2d, 2e, and 2f, respectively, show the experimentally measured transfer characteristics of the Gaussian synapse corresponding to $V_D = 0.2\text{V}$, 0.6V and 0.8V for different applied V_N .” The value of V_D is inconsistent.

We would like to thank the reviewer for pointing out this error. Note that Fig. 2 has been revised to include the characteristics of both MoS_2 and BP dual-gated (DG) FETs as well as Gaussian synapse based on their series connection in order to address the comments received from the 2nd reviewer. DGFETs allow better reconfigurability for the Gaussian synapses since the amplitude, mean and standard deviation can be independently controlled by adjusting the drain bias V_D and top-gate voltages V_N and V_P applied to the n-type MoS_2 and p-type BP FETs, respectively.

Figure 2 was revised in the manuscript.

Reviewer #2 (Remarks to the Author)

The authors report a new type of analog devices named as Gaussian synapses, based on the heterostructure of atomically thin 2D layered semiconductors, which enable the direct hardware realization of probabilistic neural networks (PNNs). They also demonstrate that PNN architecture based on Gaussian synapses is capable of recognizing complex neural oscillations or brainwave patterns from large volumes of Electroencephalography (EEG) data with extreme energy efficiency. The work is interesting and the experimental demonstration of Gaussian synapses with the heterogeneous integration of n-type MoS₂ and p-type black phosphorus (BP) based transistors is ingenious. The reviewer suggests major revisions before acceptance of the manuscript, with the following comments to be addressed:

We are happy to learn that the reviewer finds our work interesting and ingenious. We would also like to thank the reviewer for the detailed review of this paper. We have addressed the concerns and comments raised by the reviewer in a point-by-point manner. Reviewer's comments have helped us greatly to improve the quality of the revised manuscript. We hope that the reviewer finds these revisions satisfactory for acceptance of the manuscript.

- 1. The experimental demonstration of Gaussian synapses based on atomically thin 2D layered semiconductors is the cornerstone of this work. However, this manuscript lacks quantitative experimental characterization of both materials and devices. Such as thickness, size, and lattice parameters of 2D materials, the channel W/L of the devices, et al.**

We would like to thank the reviewer for pointing out to the missing information.

We have included the following information in the experimental method section:

The thicknesses of the MoS₂ and BP flakes were in the range of 3-20 nm. The thickness of the HSQ layer, used as the top-gate dielectric was 120 nm. Both devices were fabricated with a channel length of 1 μm. The width of the MoS₂ and BP devices were 0.78 μm and 2 μm, respectively. MoS₂ is a 2D layered material with the lattice parameters $a = 3.15\text{\AA}$, $b = 3.15\text{\AA}$, $c = 12.3\text{\AA}$, $\alpha =$

90° , $\beta = 90^\circ$ and $\gamma = 120^\circ$. The layered nature due to van der Waals (vdW) bonding results in a higher value for c . This enables mechanical exfoliation of the material to obtain ultra-thin layers of MoS₂. BP exhibits a puckered honeycomb lattice structure. It has phosphorous atoms existing on two parallel planes. The lattice parameters are given by $a = 3.31\text{\AA}$, $b = 10.47\text{\AA}$, $c = 4.37\text{\AA}$, $\alpha = 90^\circ$, $\beta = 90^\circ$ and $\gamma = 90^\circ$.

2. Black phosphorus is expected to oxidize upon exposure to air. The formation of surface oxide species may cause the increase in surface roughness and non-uniform degradation, thus reduced field-effect transistor performance. How to avoid the structural degradation and its effect on the electronic properties when fabricate the top-gated devices?

Reviewer's concern regarding the stability of BP and its influence on device performance is valid. Indeed BP is subject to oxidization upon exposure to air. However, this typically takes few days. The degradation is very mild for the first 5 days of ambient exposure following which the degradation increases almost exponentially and saturates after 10-11 days [15]. Initial works speculated H₂O and O₂ as the major oxidation agents responsible for oxidation of BP. Later it was found that the presence of O₂ is the major factor resulting in the oxidation of BP. This was confirmed by demonstrating the stability of BP in deoxygenated water [16]. There are various reports where the encapsulation of BP by materials such as Al₂O₃ and h-BN has improved the stability of BP and resulted in an extended life span of up to 8 months [17, 18]. In our previous submission, the BP FETs were not top-gated and as such, we had to perform all device measurements within a few days of fabricating the Gaussian synapses. However, we greatly appreciate reviewer's suggestion to demonstration dual-gate (DG) BP FETs, which allowed us to encapsulate the BP using the top-gate. We also took special care to ensure minimal exposure time to the air while fabricating BP devices by storing the material in vacuum chambers between different fabrication steps. Additionally, the three lithography steps involved in the device fabrication were done within a period of 3 days. The major part of air exposure is during the flake identification after the exfoliation (~2 hrs), following which a stack of MMA/PMMA is immediately spin coated and the source/drain contacts are defined using first electron beam (e-beam) lithography step. After lift-off, the sample is immediately loaded in a high vacuum Lake

Shore probe station and pumped down to 10^{-5} Torr. Following the measurements of the back-gate BP FETs, HSQ is spun on top of the device to act as the top-gate dielectric.

We have added the following sentences in the experimental method section:

Given the instability of BP, we took special care to ensure minimal exposure time to the air while fabricating BP devices by storing the material in vacuum chambers between different fabrication steps. Additionally, all the three lithography steps involved in the device fabrication were done within a period of 3 days.

3. The authors mentioned that their choice of MoS₂ and BP was due to their highest n-type and p-type performing. However, the carrier mobility of the n-type and p-type transistor were not balanced. Could the authors comment how this affects the reconfigurability of the Gaussian synapse? Could they design further experiments for the device optimization?

The reviewer has made an excellent observation. Indeed, the carrier mobility of the n-type and p-type transistor were not balanced. In fact, it is not necessary to have same carrier mobility for the n-type and p-type FETs since the Gaussian synapse does not utilize the ON state FET performance. Instead, the Gaussian synapse exploits the sub-threshold FET characteristics, where the slope is independent of the carrier mobility of the semiconducting channel material. For symmetric Gaussian synapse, it is, therefore, more desirable to ensure similar sub-threshold slope (SS) for the respective FETs than the carrier mobility. Ideally, SS values for both FETs should have been 60mV/decade. However, presence of interface trap capacitance worsen the SS following the equation:

$$SS = \left(\frac{k_B T}{q} \ln 10 \right) \left(1 + \frac{C_{IT}}{C_{OX}} \right) = 60 \left(1 + \frac{C_{IT}}{C_{OX}} \right) \frac{mV}{decade}; \text{ for } T = 300K$$

Where, C_{IT} and C_{OX} are, respectively, the interface trap capacitance and back-gate oxide capacitance. The SS values were found to be 4.2V/decade and 6.4 V/decade for the MoS₂ and BP FETs, respectively. Since the back-gate-oxide is the same for both FETs, i.e. 285nm SiO₂, the variation is due to C_{IT} . This is also reflected in slight asymmetry in the experimental Gaussian characteristics. SS can be improved by minimizing interface states at the 2D/gate-dielectric

interface as well as by scaling the thickness of the gate dielectric. The reason we mention high performance n-type MoS₂ and p-type BP FET has to do with the fact that metal typically form near ideal Ohmic contacts with MoS₂ and BP owing to the pinning of metal Fermi level close to the conduction and valence band, respectively [8, 19, 20]. Ohmic contact ensures that the SS is determined by the thermionic emission of carrier that follows Maxwell Boltzmann distribution and hence reaches the minimum value of 60 mV/decade at room temperature. For Schottky contact FETs, the SS value can be severely degraded due to tunneling of carriers through the Schottky barrier.

In order to avoid confusion for the readers, we have removed the phrase “Note that our choice of two different 2D materials, namely MoS₂ and BP, was motivated by the fact that these are the highest performing n-type and p-type 2D materials, respectively”. We have also added the following discussion in the revised manuscript regarding the desirable features for the Gaussian synapse:

It must be noted that the Gaussian synapses do not utilize the ON state FET performance and, therefore, are minimally influenced by the carrier mobility values of the n-type and p-type FETs. Instead, the Gaussian synapse exploits the sub-threshold FET characteristics, where the slope is independent of the carrier mobility of the semiconducting channel material. For symmetric Gaussian synapse, it is, therefore, more desirable to ensure similar sub-threshold slope (SS) for the respective FETs than the carrier mobility. Ideally, SS values for both FETs should be 60mV/decade. However, presence of interface trap capacitance worsen the SS. SS can be improved by minimizing interface states at the 2D/gate-dielectric interface as well as by scaling the thickness of the gate dielectric. It is also desirable to have FETs with Ohmic contacts for Gaussian synapses to ensure that the SS is determined by the thermionic emission of carriers to reach the minimum value of 60 mV/decade at room temperature. For Schottky contact FETs, the SS can be severely degraded due to tunneling of carriers through the Schottky barrier.

4. By top gating the MoS₂ FET, the device transfer function can be tuned. While the reconfigurability of the Gaussian synapses can be further extended if the BP FET is also top-gated. The authors should provide the transfer characteristics of the BP DGFET.

We agree with the reviewer that the transfer characteristics of BP DGFET should be provided for complete reconfigurability of the Gaussian synapses. We have revised Fig. 2 to include the characteristics of both MoS₂ and BP dual-gated (DG) FETs as well as Gaussian synapse based on their series connection. Fig. 2 is also included in this document as Fig R2.

Figure R2. Reconfigurable Gaussian synapse. a) Schematic of a reconfigurable Gaussian synapse involving dual-gated n-type MoS₂ FET and p-type BP FET. The top-gate stack was fabricated using hydrogen silsesquioxane (HSQ) as the top-gate dielectric and nickel/gold (Ni/Au) as the top-gate electrode. b) Back-gated transfer characteristics of the MoS₂ FET at $V_D = 1V$, for different top-gate voltages (V_N). c) Back-gate threshold voltage, V_{TN} of MoS₂ FET as function of V_N was extracted using the constant current method. Inset shows the band diagram elucidating how V_N controls V_{TN} by electrostatically adjusting the height of the thermal energy barrier for electron injection into the MoS₂ channel. d) Back-gated transfer characteristics of the BP FET at $V_D = 1V$, for different top-gate voltages (V_P). e) Back-gate threshold voltage, V_{TP} of BP FET as function of V_P . Inset shows the band diagram elucidating how V_P controls V_{TP} by electrostatically adjusting the height of the thermal energy barrier for hole injection into the BP channel. As expected the slope ($\alpha_N = 1.91$) of V_{TN} versus V_N and the slope ($\alpha_P = 2$) of V_{TP} versus V_P are found to be similar to the ratio of top-gate capacitance (C_{TG}) to the back-gate capacitance (C_{BG}), which in our case was ~ 1.94 , given that the top-gate and back-gate dielectric thicknesses are 120 nm and 285 nm, and dielectric constant are ~ 3.2 and ~ 3.9 , respectively. f) Transfer characteristics of the Gaussian synapse for different values of $V_N = V_P$. This configuration allows us to shift the mean (μ_V) of the Gaussian synapse without changing the amplitude (A) and the standard deviation (σ_V). g) μ_V as a function of $V_N = V_P$. h) Transfer characteristics of the Gaussian synapse for different values of $V_N = -V_P$. This configuration allows us to configure σ_V while keeping μ_V constant. i) σ_V as a function of $V_N = -V_P$. However, this configuration also results in an increase in the amplitude (A) of the Gaussian synapse as σ_V increases. This increase can be adjusted by changing the drain voltage (V_D) since A is linearly proportional to V_D . Nevertheless, by controlling V_N, V_P and V_D , it is possible to adjust the mean, standard deviation and amplitude of the Gaussian synapse.

We have revised Fig. 2 to include the characteristics of both MoS₂ and BP dual-gated (DG) FETs as well as Gaussian synapse based on their series connection. The following discussion was also added:

Fig. 2a shows the schematic representation of a reconfigurable Gaussian synapse, where, both MoS₂ and BP FETs are dual-gated (DG). The top-gate stack was fabricated using hydrogen silsesquioxane (HSQ)[21] as the top-gate dielectric and nickel/gold (Ni/Au) as the top-gate electrode. The fabrication process flow is described in the experimental method section. Fig. 2b shows the experimentally measured back-gated transfer characteristics of the MoS₂ FET at $V_D = 1\text{V}$, for different top-gate voltages (V_N). Clearly, V_N controls the back-gate threshold voltage, V_{TN} of the MoS₂ FET as shown in Fig. 2c. The energy band diagram shown in the inset of Fig. 2c can be used to explain the concept of threshold voltage engineering using gate electrostatics. The top-gate voltage determines the height of the potential barrier for electron injection inside the MoS₂ channel, which must be overcome by applying back-gate voltage to enable current conduction from the source to the drain terminal. Negative top-gate voltages increase the potential barrier for electron injection and hence necessitate larger positive back-gate voltages to achieve similar level of current conduction. As such, V_{TN} becomes more positive (less negative) for large negative V_N values. Note that the slope (α_N) of V_{TN} versus V_N in Fig. 2c must be proportional to the ratio of top-gate capacitance (C_{TG}) to the back-gate capacitance (C_{BG}). This follows directly from the principle of charge balance, which ensures that the channel charge induced by the top-gate voltage must be compensated by the back-gate voltage at threshold. We extracted the value for α_N to be ~ 1.91 . This is consistent with the theoretical prediction of ~ 1.94 , given that the top-gate and back-gate dielectric thicknesses are 120 nm and 285 nm, respectively and the top-gate insulator, HSQ, has a slightly lower dielectric constant (~ 3.2) than the back-gate insulator, SiO₂ (3.9). Fig. 2d shows the experimentally measured back-gated transfer characteristics of the BP FET at $V_D = 1\text{V}$, for different top-gate voltages (V_P). As expected, V_P controls the back-gate threshold voltage, V_{TP} of the BP FET as shown in Fig. 2e. A similar explanation follows except for the fact that the top-gate voltage in case of BP FET influences the height of the potential barrier for hole injection, which is overcome by applying back-gate voltage enabling current conduction from the drain to the source terminal. The corresponding energy band diagram is shown in the inset of Fig. 2e. Positive top-gate voltages increase the potential barrier for hole injection and hence necessitate larger negative back-gate voltages to achieve similar level of current conduction. As such, V_{TP}

becomes more negative (less positive) for large positive V_P values. We also extracted the slope (α_P) of V_{TP} versus V_P in Fig. 2e and, as expected, found a similar value of ~ 2 .

The dual-gated MoS₂ and BP FETs allow complete control of the shape of the Gaussian synapse. Fig. 2f shows the experimentally measured transfer characteristics of the Gaussian synapse for different values of $V_N = V_P$. This configuration allows us to shift the mean (μ_V) of the Gaussian synapse without changing the amplitude (A) and the standard deviation (σ_V) Fig. 2g shows μ_V plotted as a function of $V_N = V_P$. We are able to do this since the back-gate threshold voltage for both MoS₂ and BP FETs shift in the same direction in this configuration. Similarly, Fig. 2h shows the experimentally measured transfer characteristics of the Gaussian synapse for different values of $V_N = -V_P$. Under this configuration, the back-gate threshold voltage for MoS₂ and BP FETs shift in the opposite direction. As such the μ_V of the Gaussian distribution remains constant, whereas, σ_V keeps increasing. Fig. 2i shows σ_V plotted as a function of $V_N = -V_P$. However, this configuration also results in an increase in the amplitude (A) of the Gaussian synapse as σ_V increases. This increase can be adjusted by changing the drain voltage (V_D) since A is linearly proportional to V_D . Nevertheless, by controlling V_N, V_P and V_D , it is possible to adjust the mean, standard deviation and amplitude of the Gaussian synapse.

5. The authors should explain clearly the working principle of the Gaussian synapse and the role of the Common terminal in the Fig. 1e and Fig. 2a.

We are happy to explain the working principle of the Gaussian synapse. First, we would like to clarify that the function of the common terminal is simply to connect the MoS₂ and BP FETs together in series. This terminal need not be probe, when implemented into integrated circuits. However, from an academic point a view the common terminal allows us to measure the MoS₂ and BP FETs individually and obtain their respective transfer characteristics.

The working principle of Gaussian synapse can be explained seamlessly by acknowledging the fact that FETs are essentially gate-controlled variable resistors. As such, the Gaussian synapse can be represented by a network of two variable resistors connected in series as shown in Fig. R3a. The two variable resistors, i.e. R_{MoS_2} and R_{BP} correspond to the MoS₂ and BP FETs, whose transfer characteristics are shown in Fig. R3c and R3d, respectively. The current (I_D) through the Gaussian synapse is given by the following expression:

$$I_D = \frac{V_D}{R_{MoS_2} + R_{BP}}$$

For large negative V_{BG} beyond $\approx -30V$ (denoted in purple), the MoS_2 FET is in the OFF state with $R_{MoS_2} \approx 10^{12}\Omega$, and the BP FET is in the ON state with $R_{BP} \approx 10^3\Omega$. Since the resistors are connected in series, the higher resistance dominates and prevents any current conduction through the Gaussian synapse. We see a similar situation for large positive V_{BG} beyond $\approx 30V$ (denoted in red). In this case, the MoS_2 FET is in the ON state with $R_{MoS_2} \approx 10^3\Omega$, and the BP FET is in the OFF state with $R_{BP} \approx 10^{12}\Omega$. However, as we change the V_{BG} from $-30V$ to $30V$, first, the current conduction through the Gaussian synapse increases exponentially following the subthreshold characteristics of the MoS_2 FET. The current reaches its peak magnitude at the V_{BG} value for which $R_{MoS_2} = R_{BP}$ (denoted in green) and its magnitude is proportional to the supply voltage (V_D). Beyond this peak, current conduction is dominated by the exponential subthreshold characteristics of the BP FET. As a result, the MoS_2/BP heterostructure exhibits non-monotonic transfer characteristics with exponential tails and mimics Gaussian distribution.

Figure R3. Principle of operation of Gaussian synapse. a) Schematic, equivalent circuit diagram and transfer characteristics of the Gaussian synapse for different supply voltage (V_D). b) Transfer characteristics of BP FET. c) Transfer characteristics of MoS_2 FET. Colored circles represent different regions of operation of the Gaussian synapse, which is represented by a network of two gate controlled variable resistors connected in series.

We have added the equivalent circuit diagram consisting of two variable resistors in Fig. 1e of the revised manuscript. We have also added the following discussion in the revised manuscript:

Fig. 1e also shows the equivalent circuit diagram for the Gaussian synapse, which simply consists of two variable resistors in series. The two variable resistors, i.e. R_{MoS_2} and R_{BP} correspond to the MoS_2 and BP FETs. From the equivalent circuit diagram, the current (I_D) through the Gaussian synapse can be written as:

$$I_D = \frac{V_D}{R_{MoS_2} + R_{BP}}$$

For extreme V_{BG} values, i.e. large negative ($< -30V$) and large positive ($> 30V$), the MoS₂ FET and the BP FET are in their respective OFF states, making the corresponding resistances, i.e. R_{MoS_2} and R_{BP} very large (approximately T Ω). This prevents any current conduction between the source and the drain terminal of the Gaussian synapse. However, as the MoS₂ FET switches from OFF state to ON state, current conduction begins and increases exponentially with V_{BG} following the subthreshold characteristics and reaches its peak magnitude determined by the V_D . Beyond this peak, the current starts to decrease exponentially following the subthreshold characteristics of the BP FET. As a result, the series connection of MoS₂ and BP FETs exhibits non-monotonic transfer characteristics with exponential tails that mimics Gaussian distribution.

6. Fig 2 (b) shown the transfer characteristics of MoS₂ DGFET for different top-gate voltages (V_N), the authors should explain this phenomenon from the energy level, interface analysis and add more discussions about the mechanisms.

We have revised Fig. 2 to include the characteristics of both MoS₂ and BP dual-gated (DG) FETs as well as Gaussian synapse based on their series connection. We have explained the working principle of MoS₂ and BP DGFETs accordingly in the revised manuscript:

Fig. 2b shows the experimentally measured back-gated transfer characteristics of the MoS₂ FET at $V_D = 1V$, for different top-gate voltages (V_N). Clearly, V_N controls the back-gate threshold voltage, V_{TN} of the MoS₂ FET as shown in Fig. 2c. The energy band diagram shown in the inset of Fig. 2c can be used to explain the concept of threshold voltage engineering using gate electrostatics. The top-gate voltage determines the height of the potential barrier for electron injection inside the MoS₂ channel, which must be overcome by applying back-gate voltage to enable current conduction from the source to the drain terminal. Negative top-gate voltages increase the potential barrier for electron injection and hence necessitate larger positive back-gate voltages to achieve similar level of current conduction. As such, V_{TN} becomes more positive (less negative) for large negative V_N values. Note that the slope (α_N) of V_{TN} versus V_N in Fig. 2c must be proportional to the ratio of back-gate capacitance (C_{BG}) to the top-gate capacitance (C_{TG}). This follows directly from the principle of charge balance, which ensures that the channel charge induced by the top-gate voltage must be compensated by the back-gate voltage at threshold. We extracted the value for α_N to be ~ 1.91 . This is consistent with the theoretical prediction of ~ 1.94 ,

given that the top-gate and back-gate dielectric thicknesses are 120 nm and 285 nm, respectively and the top-gate insulator, HSQ, has a slightly lower dielectric constant (~ 3.2) than the back-gate insulator, SiO₂ (3.9). Fig. 2d shows the experimentally measured back-gated transfer characteristics of the BP FET at $V_D = 1V$, for different top-gate voltages (V_P). As expected, V_P controls the back-gate threshold voltage, V_{TP} of the BP FET as shown in Fig. 2e. A similar explanation follows except for the fact that the top-gate voltage in this case determines the height of the potential barrier for hole injection inside the BP channel, which must be overcome by applying back-gate voltage to enable current conduction from the drain to the source terminal. The corresponding energy band diagram is shown in the inset of Fig. 2e. Positive top-gate voltages increase the potential barrier for hole injection and hence necessitate larger negative back-gate voltages to achieve similar level of current conduction. As such, V_{TP} becomes more negative (less positive) for large positive V_P values. We also extracted the slope (α_P) of V_{TP} versus V_P in Fig. 2e and, as expected, found a similar value of ~ 2 .

7. The authors claims the slope (α) of VT versus VN in Fig. 2c is proportional to the ratio of top-gate capacitance (CTG) to the back-gate capacitance (CBG), which was found to be $\sim 2/3$. It lacks adequate evidence. VT versus VN is related to the interface between MoS₂ and HSQ or the energy level of MoS₂, when the interface is different, the charge trapped capability is different and the ratio of VT and VN will be different. So, the authors need to further study the interface between MoS₂ and HSQ. In other words, only when the interface of MoS₂ and HSQ is the same as the interface of MoS₂ and SiO₂, the slope (α) of VT versus VN may compare with the ratio of top-gate capacitance (CTG) to the back-gate capacitance (CBG).

The reviewer has brought up an excellent point. We do agree that it is important to characterize the interface between the semiconductor and the top and bottom gate dielectric for a better understanding of the threshold engineering through electrostatic gating. Before we address this comment, we would like to point out that Fig. 2 was revised to include the characteristics of both MoS₂ and BP dual-gated (DG) FETs as well as Gaussian synapse based on their series connection in accordance with your earlier request. We also switched to 285nm SiO₂ as the back-gate

dielectric due to non-academic reasons. The back-gate oxide thickness does not alter any scientific findings except for the use of slightly higher back-gate voltages for the device measurements.

The reviewer is correct in the assessment that the slope, α_N of V_{TN} versus V_N for the MoS₂ DGFET will be equal to the ratio of top-gate capacitance (C_{TG}) to the back-gate capacitance (C_{BG}) only when the interface of MoS₂/HSQ is the same as the interface of MoS₂/SiO₂. This is indeed what we found following additional experiments to characterize these interfaces through hysteresis measurements. Fig. R4a and Fig. R4b show the dual sweep for the back-gate and top-gate transfer characteristics of MoS₂ DGFET with corresponding hysteresis windows of $\sim 3V$ and $\sim 2.5V$, respectively. Since hysteresis window is a measure for the degree of trapping at the 2D/gate-dielectric interface, it is fair to assume that the two interfaces are very similar in nature. Furthermore, similar values of constant-current threshold voltages were extracted from both forward and reverse sweep of the back-gate and top-gate characteristics that reinforces our claim.

Figure R4. Dielectric Interface Characterization for MoS₂ DGFET. Dual sweep for a) back-gate and b) top-gate transfer characteristics of MoS₂ DGFET. Table shows that the corresponding hysteresis windows are $\sim 3V$ and $\sim 2.5V$, respectively. Since hysteresis window is a measure for the degree of trapping at the 2D/gate-dielectric interface, it is fair to assume that the two interfaces are very similar in nature. Furthermore, similar values of constant-current threshold voltages were extracted from both forward and reverse sweep of the back-gate and top-gate characteristics.

We have included this dielectric interface characterization for MoS₂ DGFET in the supporting information section.

8. The authors claimed that the PNN architectures based on Gaussian synapses could offer extreme energy efficiency. They use a Virtual Source (VS) model to project the performance of scaled Gaussian synapse. However, it is not guaranteed that a scaled device will show similar behaviors with a lowered energy consumption. Also, the authors

need to comment on the energy consumption and compare them to literature to highlight the advantage of this work.

Reviewer's comment regarding the validity of scaling is interesting. However, please note that voltage scaling is a relatively well understood and time tested approach used by the semiconductor industry for nearly four decades (1965-2005) to reduce the power dissipation of transistors. The dynamic power dissipation ($P_{dynamic} = \eta CV_{DD}^2 f$) by a transistor is attributed to state switching, which is proportional to the square of the supply voltage (V_{DD}), with η as the activity factor, C as the capacitance of the circuit, and f is the switching frequency. During the Golden era of MOSFET scaling, also referred to as the Dennard scaling era, the power consumption of the chip was kept practically constant by scaling the threshold voltage (V_{TH}) and concurrently the supply voltage (V_{DD}) of the MOSFET. However, in 2005 the voltage scaling stalled since further reduction in V_{TH} resulted in an exponential increase in the static leakage current (I_{static}) and hence static power consumption ($P_{static} = I_{static}V_{DD}$) as shown in Fig. R5a. This is a direct consequence of non-scalability of the subthreshold swing (SS) to below 60mV/decade also known as the Boltzmann tyranny. In fact the total power consumption (P_{total}) by any integrated circuits (ICs) of the present Dark-Si era is mostly dominated by the static power dissipation ($P_{total} = P_{static} + P_{dynamic}$) compared to the dynamic switching power. Nevertheless, reinstating V_{DD} scaling can help us to escape the Boltzmann tyranny by minimizing both the static as well as dynamic power consumption. Therefore, in recent years, subthreshold logic circuits, which utilize supply voltage (V_{DD}) that is close to or even less than V_{TH} , have received significant attention for ultra-low power applications [22, 23]. New subthreshold logic and memory design methodologies have already been developed and demonstrated on a fast Fourier transform (FFT) processor [24] and analog VLSI neural systems [25].

Note that our proposed Gaussian synapses naturally require operation in subthreshold regime in order to exploit the exponential feature in the transfer characteristics of the n-type and p-type transistors. Furthermore, as shown in Fig. R5b, Gaussian synapses maintain their characteristics features even when the supply voltage (V_{DD}) is scaled down to 200 mV. This allows the Gaussian synapses to be inherently low power. For the proof-of-concept demonstration of Gaussian synapses, we have used relatively thicker back-gate and top-gate oxides with respective thicknesses of 285 nm and 120 nm. This necessitates the use of rather large back-gate and top-gate

voltages in the range of -50 V to 50V and -35 V to 35 V, respectively. The power consumption by our proof-of-concept Gaussian synapses will still be high in spite of scaling the supply voltage and exploiting the subthreshold current conduction in the range of nano amperes through the MoS₂ and BP FETs. This is because power consumption by Gaussian synapse is simply the area under the I_D versus V_{BG} curve. By scaling the thicknesses of both the top and bottom gate dielectrics, it is possible to scale the operating gate

Figure R5. Overcoming Boltzmann Tyranny. a) Non-scalability of sub-threshold slope to below 60 mV/decade in MOSFETs restrict the scaling of supply voltage (V_{DD}). b) Gaussian synapses maintain their characteristics features even when V_{DD} is scaled down to 200 mV or beyond and are not limited by supply voltage scaling, which allow them to operate with low power compared to MOSFETs.

voltages and thereby archive the desirable power benefits from the Gaussian synapses. Ultra-thin dielectric materials such as Al₂O₃ and HfO₂, which offer much larger dielectric constants, ϵ_{ox} , ~9 and ~25 respectively, are now routinely used as gate oxides for highly scaled Si FinFETs [26].

We have included the following discussion in the revised manuscript:

Note that our proposed Gaussian synapses naturally require operation in subthreshold regime in order to exploit the exponential feature in the transfer characteristics of the n-type and p-type transistors. Furthermore, as shown in Fig. 1f, Gaussian synapses maintain their characteristics features even when the supply voltage (V_{DS}) is scaled down to 200 mV or even beyond. This allows the Gaussian synapses to be inherently low power. For the proof-of-concept demonstration of Gaussian synapses, we have used relatively thicker back-gate and top-gate oxides with respective thicknesses of 285 nm and 120 nm. This necessitates the use of rather large back-gate and top-gate voltages in the range of -50 V to 50V and -35 V to 35 V, respectively. The power consumption by our proof-of-concept Gaussian synapses will still be high in spite of scaling the supply voltage and exploiting the subthreshold current conduction in the range of nano amperes through the MoS₂ and BP FETs. This is because power consumption by Gaussian synapse is simply the area under the I_D versus V_{BG} curve. By scaling the thicknesses of both the top and bottom gate dielectrics, it is possible to scale the operating gate voltages and thereby achieve desirable power benefits from the Gaussian synapses. Ultra-thin dielectric materials such as Al₂O₃ and HfO₂, which offer much

larger dielectric constants, ϵ_{ox} , ~ 9 and ~ 25 respectively, are now routinely used as gate oxides for highly scaled Si FinFETs.[26]

The reviewer has also asked for a comparison of energy consumption by the PNN based on Gaussian synapses against literature to highlight the advantage of this work.

Recently, Saleheen, M. et al., have used field-programmable gate array (FPGA) for developing a real-time seizure detector using multi-layer perceptron (MLP) based artificial neural networks (ANN) with an accuracy of 99.18% and power consumption on the order of 1.5 mW [13]. Similarly, Atlaf, M. A. B. et al. demonstrated a rapid-eye movement (REM) versus non-REM sleep classifier for Alzheimer patients with a sensitivity of 89.8% and specificity of 93.6% using a linear SVM classifier on FPGA platform [14]. Low-power classification was obtained with a power consumption of 20.8 μ W. With our PNN architecture, we are able to calculate the percentage of different rhythmic activity such as δ , θ , α , β and γ waves, which can determine the physical state of a person. The PNN architecture can certainly be extended to create a classifier, which could use the percentage strength of different EEG oscillations to predict the physical state of a person. Since the function of our PNN architecture is to find the percentage strength of different brainwaves, it would not be ideal to compare it to the classifiers described above, which are also used in different contexts. Nevertheless, even for a large sample volume of $N = 2 \times 10^5$, corresponding to 8 hrs of EEG data, the power consumption by our PNN architecture was found to be as frugal as 3 μ W for $V_{DD} = 0.1V$, which increases to only 350 μ W for $V_{DD} = 1.0V$.

9. The experimental results were all obtained from single devices. However, the realization of PNN in more complicated programs require more Gaussian synapses. The device variation is a critical issue for the achievement of a large-size array. Was the reliability and uniformity issue of the devices?

This is an excellent point raised by the reviewer. We agree that device-to-device variation must be eliminated before Gaussian synapses can be used for large-scale implementation of circuits. We measured the back-gated transfer characteristics for 14 MoS₂ and 16 BP DGFETs as shown in Fig. R6a, and Fig. R6b, respectively. Clearly, there exists significant variation in the device characteristics. The variation is mostly attributed to the thicknesses of the corresponding MoS₂

and BP flakes obtained through the random exfoliation process. The greater variation in BP can be attributed to the layer-number dependent bandgap for the material [27]. This variability can be eliminated by using large-area grown 2D materials with uniform thickness. Fortunately, recent years have seen tremendous progress in wafer-scale growth of high quality MoS₂ and BP using chemical vapor deposition (CVD) and metal organic chemical vapor deposition (MOCVD) techniques [1-5, 28]. Furthermore, while we have used MoS₂ and BP for our proof-of-concept demonstration of Gaussian synapses, there are 2D materials, such as, WSe₂, which offer ambipolar transport, i.e. the presence of both electron and hole conduction [6] and can be grown over large area using CVD techniques.

However, the performance of WSe₂ based n-type and p-type FETs are limited by the presence of large Schottky barriers at the metal/2D contact interfaces. By resolving the contact resistance related issues and improving the quality of large area synthesized WSe₂, it is

Figure R6. Device-to-device Variation. Back-gated transfer characteristics of a) 14 MoS₂ and b) 16 BP DGFETs. The variation is mostly attributed to the thicknesses of the corresponding MoS₂ and BP flakes obtained through the random exfoliation process. The greater variation in BP can be attributed to the layer-number dependent bandgap for the material. This variability can be eliminated by using large area grown 2D materials with uniform thickness.

possible to implement Gaussian synapses based on WSe₂ as well. Moreover, in recent years several groups have demonstrated p-type MoS₂ and n-type BP, by implementing smart contact engineering and/or doping strategies [9, 10]. Therefore, very large scale integration of Gaussian synapses based on CVD grown single 2D material will be possible in the near future for the hardware realization of PNNs. Since the focus of this article is to introduce the novel Gaussian synapse and its benefit as a statistical computing primitive, material optimization is beyond the scope and will be investigated in our future studies.

10. On the Brainwave Recognition : the authors state that the PNN recognizes the dominant presence of delta and theta waves in the sleep samples, almost with 100% accuracy. How about the accuracy when conceive of much more complex sample selection?

This is an excellent question. With our PNN architecture, we are able to calculate the percentage of different rhythmic activity such as δ , θ , α , β and γ waves, which can determine the physical state of a person. As shown in Fig. R1 the PNN identifies higher percentage of low-frequency δ waves in the EEG samples corresponding to the sleep data, whereas, the awake state data has higher percentage of θ and α waves. The PNN architecture can certainly be extended to create a classifier, which could use the percentage strength of different EEG oscillations to predict the physical state of a person. We would investigate more complex brainwave patterns in our future studies.

11. There are many format errors in the text and references. For example, ref 1/ref 9, ref 8/ref 63 is the same. There are some typos and grammatical mistakes in the paper, for example, “mersister” on the page 4 line 6.

These have been fixed in the manuscript.

Reviewer #3 (Remarks to the Author):

This paper describes a new electronic synapse created as a combination of n-type and p-type back-gate field-effect transistors in a single integrated device, including means to tune the resulting Gaussian-like characteristic. Experimental measurements of the device are included, and a model is then simulated to suggest classification of brain-wave data.

The contribution in device development may be significant but the application to brain-wave classification is ad hoc, and its demonstration far-fetched. The Fourier transforms of EEG synthesized data are computed and then the simulated spectra are presented as voltage to a collection of simulated synapses to implement a Gaussian mixture model.

In all, this paper makes bold claims on a demonstration of neuromorphic computing beating Boltzmann and von Neumann limits and achieving brain-wave detection that are not substantiated. A much more concrete and focused demonstration of the device electronic capabilities would serve the reader much better than the random jumble of deep learning and thermodynamic jargon without substance.

We would like to thank the reviewer for acknowledging the fact that the contribution of this manuscript towards the development of a novel device is significant. We also appreciate the reviewer's suggestion to focus more on the demonstration of the electronic capabilities of the device. You will find that in response to the specific comments made by the two other reviewers, we have now included additional experimental data and more in depth analysis of the Gaussian synapse. We have also made significant effort to substantiate on the low power aspect of the Gaussian synapse.

Some example changes are listed below:

1. We have added the equivalent circuit diagram consisting of two variable resistors in Fig. 1e of the revised manuscript. We have also added the following discussion in the revised manuscript: Fig. 1e also shows the equivalent circuit diagram for the Gaussian synapse, which simply consists of two variable resistors in series. The two variable resistors, i.e. R_{MoS_2} and R_{BP} correspond to the MoS₂ and BP FETs. From the equivalent circuit diagram, the current (I_D) through the Gaussian synapse can be written as:

$$I_D = \frac{V_D}{R_{MoS_2} + R_{BP}}$$

For extreme V_{BG} values, i.e. large negative ($< -30V$) and large positive ($> 30V$), the MoS₂ FET and the BP FET are in their respective OFF states, making the corresponding resistances, i.e. R_{MoS_2} and R_{BP} very large (approximately TΩ). This prevents any current conduction between the source and the drain terminal of the Gaussian synapse. However, as the MoS₂ FET switches from OFF state to ON state, current conduction begins and increases exponentially with V_{BG} following the subthreshold characteristics and reaches its peak magnitude determined by the V_D . Beyond this peak, the current starts to decrease exponentially following the subthreshold characteristics of the BP FET. As a result, the series connection of MoS₂ and BP FETs exhibits non-monotonic transfer characteristics with exponential tails that mimics Gaussian distribution.

2. We have also added the following discussion in the revised manuscript regarding the desirable features for the Gaussian synapse:

It must be noted that the Gaussian synapses do not utilize the ON state FET performance and, therefore, are minimally influenced by the carrier mobility values of the n-type and p-type FETs. Instead, the Gaussian synapse exploits the sub-threshold FET characteristics, where the slope is independent of the carrier mobility of the semiconducting channel material. For symmetric Gaussian synapse, it is, therefore, more desirable to ensure similar sub-threshold slope (SS) for the respective FETs than the carrier mobility. Ideally, SS values for both FETs should be 60mV/decade. However, presence of interface trap capacitance worsen the SS. SS can be improved by minimizing interface states at the 2D/gate-dielectric interface as well as by scaling the thickness of the gate dielectric. It is also desirable to have Ohmic contact FETs for Gaussian synapses to ensure that the SS is determined by the thermionic emission of carriers and hence reaches the minimum value of 60 mV/decade at room temperature. For Schottky contact FETs, the SS value can be severely degraded due to tunneling of carriers through the Schottky barrier.

3. We have revised Fig. 2 to include the characteristics of both MoS₂ and BP dual-gated (DG) FETs as well as Gaussian synapse based on their series connection. The following discussion was also added:

Fig. 2a shows the schematic representation of a reconfigurable Gaussian synapse, where, both MoS₂ and BP FETs are dual-gated (DG). The top-gate stack was fabricated using hydrogen silsesquioxane (HSQ)[21] as the top-gate dielectric and nickel/gold (Ni/Au) as the top-gate electrode. The fabrication process flow is described in the experimental method section. Fig. 2b shows the experimentally measured back-gated transfer characteristics of the MoS₂ FET at $V_D = 1\text{V}$, for different top-gate voltages (V_N). Clearly, V_N controls the back-gate threshold voltage, V_{TN} of the MoS₂ FET as shown in Fig. 2c. The energy band diagram shown in the inset of Fig. 2c can be used to explain the concept of threshold voltage engineering using gate electrostatics. The top-gate voltage determines the height of the potential barrier for electron injection inside the MoS₂ channel, which must be overcome by applying back-gate voltage to enable current conduction from the source to the drain terminal. Negative top-gate voltages increase the potential barrier for electron injection and hence necessitate larger positive back-gate voltages to achieve similar level of current conduction. As such, V_{TN} becomes more positive (less negative) for large negative V_N values. Note that the slope (α_N) of V_{TN} versus V_N in Fig. 2c must be proportional to the ratio of top-gate capacitance (C_{TG}) to the back-gate capacitance (C_{BG}). This follows directly from the principle of charge balance, which ensures that the channel charge induced by the top-gate voltage must be compensated by the back-gate voltage at threshold. We extracted the value for α_N to be ~ 1.91 . This is consistent with the theoretical prediction of ~ 1.94 , given that the top-gate and back-gate dielectric thicknesses are 120 nm and 285 nm, respectively and the top-gate insulator, HSQ, has a slightly lower dielectric constant (~ 3.2) than the back-gate insulator, SiO₂ (3.9). Fig. 2d shows the experimentally measured back-gated transfer characteristics of the BP FET at $V_D = 1\text{V}$, for different top-gate voltages (V_P). As expected, V_P controls the back-gate threshold voltage, V_{TP} of the BP FET as shown in Fig. 2e. A similar explanation follows except for the fact that the top-gate voltage in case of BP FET influences the height of the potential barrier for hole injection, which is overcome by applying back-gate voltage enabling current conduction from the drain to the source terminal. The corresponding energy band diagram is shown in the inset of Fig. 2e. Positive top-gate voltages increase the potential barrier for hole injection and hence necessitate larger negative back-gate voltages to achieve similar level of current conduction. As such, V_{TP} becomes more negative (less positive) for large positive V_P values. We also extracted the slope (α_P) of V_{TP} versus V_P in Fig. 2e and, as expected, found a similar value of ~ 2 .

Figure 2. Reconfigurable Gaussian synapse. a) Schematic of a reconfigurable Gaussian synapse involving dual-gated n-type MoS₂ FET and p-type BP FET. The top-gate stack was fabricated using hydrogen silsesquioxane (HSQ) as the top-gate dielectric and nickel/gold (Ni/Au) as the top-gate electrode. b) Back-gated transfer characteristics of the MoS₂ FET at $V_D = 1V$, for different top-gate voltages (V_N). c) Back-gate threshold voltage, V_{TN} of MoS₂ FET as function of V_N was extracted using the constant current method. Inset shows the band diagram elucidating how V_N controls V_{TN} by electrostatically adjusting the height of the thermal energy barrier for electron injection into the MoS₂ channel. d) Back-gated transfer characteristics of the BP FET at $V_D = 1V$, for different top-gate voltages (V_P). e) Back-gate threshold voltage, V_{TP} of BP FET as function of V_P . Inset shows the band diagram elucidating how V_P controls V_{TP} by electrostatically adjusting the height of the thermal energy barrier for hole injection into the BP channel. As expected the slope ($\alpha_N = 1.91$) of V_{TN} versus V_N and the slope ($\alpha_P = 2$) of V_{TP} versus V_P are found to be similar to the ratio of top-gate capacitance (C_{TG}) to the back-gate capacitance (C_{BG}), which in our case was ~ 1.94 , given that the top-gate and back-gate dielectric thicknesses are 120 nm and 285 nm, and dielectric constant are ~ 3.2 and ~ 3.9 , respectively. f) Transfer characteristics of the Gaussian synapse for different values of $V_N = V_P$. This configuration allows us to shift the mean (μ_V) of the Gaussian synapse without changing the amplitude (A) and the standard deviation (σ_V). g) μ_V as a function of $V_N = V_P$. h) Transfer characteristics of the Gaussian synapse for different values of $V_N = -V_P$. This configuration allows us to configure σ_V while keeping μ_V constant. i) σ_V as a function of $V_N = -V_P$. However, this configuration also results in an increase in the amplitude (A) of the Gaussian synapse as σ_V increases. This increase can be adjusted by changing the drain voltage (V_D) since A is linearly proportional to V_D . Nevertheless, by controlling V_N, V_P and V_D , it is possible to adjust the mean, standard deviation and amplitude of the Gaussian synapse.

The dual-gated MoS₂ and BP FETs allow complete control of the shape of the Gaussian synapse.

Fig. 2f shows the experimentally measured transfer characteristics of the Gaussian synapse for different values of $V_N = V_P$. This configuration allows us to shift the mean (μ_V) of the Gaussian synapse without changing the amplitude (A) and the standard deviation (σ_V) Fig. 2g shows μ_V plotted as a function of $V_N = V_P$. We are able to do this since the back-gate threshold voltage for both MoS₂ and BP FETs shift in the same direction in this configuration. Similarly, Fig. 2h shows

the experimentally measured transfer characteristics of the Gaussian synapse for different values of $V_N = -V_P$. Under this configuration, the back-gate threshold voltage for MoS₂ and BP FETs shift in the opposite direction. As such the μ_V of the Gaussian distribution remains constant, whereas, σ_V keeps increasing. Fig. 2i shows σ_V plotted as a function of $V_N = -V_P$. However, this configuration also results in an increase in the amplitude (A) of the Gaussian synapse as σ_V increases. This increase can be adjusted by changing the drain voltage (V_D) since A is linearly proportional to V_D . Nevertheless, by controlling V_N, V_P and V_D , it is possible to adjust the mean, standard deviation and amplitude of the Gaussian synapse.

4. We have included the following discussion regarding low power aspect of the Gaussian synapse: Note that our proposed Gaussian synapses naturally require operation in subthreshold regime in order to exploit the exponential feature in the transfer characteristics of the n-type and p-type transistors. Furthermore, as shown in Fig. 1f, Gaussian synapses maintain their characteristics features even when the supply voltage (V_{DS}) is scaled down to 200 mV or even beyond. This allows the Gaussian synapses to be inherently low power. For the proof-of-concept demonstration of Gaussian synapses, we have used relatively thicker back-gate and top-gate oxides with respective thicknesses of 285 nm and 120 nm. This necessitates the use of rather large back-gate and top-gate voltages in the range of -50 V to 50V and -35 V to 35 V, respectively. The power consumption by our proof-of-concept Gaussian synapses will still be high in spite of scaling the supply voltage and exploiting the subthreshold current conduction in the range of nano amperes through the MoS₂ and BP FETs. This is because power consumption by Gaussian synapse is simply the area under the I_D *versus* V_{BG} curve. By scaling the thicknesses of both the top and bottom gate dielectrics, it is possible to scale the operating gate voltages and thereby archive desirable power benefits from the Gaussian synapses. Ultra-thin dielectric materials such as Al₂O₃ and HfO₂, which offer much larger dielectric constants, ϵ_{ox} , ~9 and ~25 respectively, are now routinely used as gate oxides for highly scaled Si FinFETs [26].

References

- [1] D. Andrzejewski, M. Marx, A. Grundmann, O. Pfingsten, H. Kalisch, A. Vescan, *et al.*, "Improved luminescence properties of MoS₂ monolayers grown via MOCVD: role of pre-treatment and growth parameters," *Nanotechnology*, vol. 29, p. 295704, Jul 20 2018.
- [2] T. K. Nguyen, A. D. Nguyen, C. T. Le, F. Ullah, Z. Tahir, K. I. Koo, *et al.*, "High Photoresponse in Conformally Grown Monolayer MoS₂ on a Rugged Substrate," *ACS Appl Mater Interfaces*, vol. 10, pp. 40824-40830, Nov 28 2018.
- [3] K. K. H. Smithe, S. V. Suryavanshi, M. Munoz Rojo, A. D. Tedjarati, and E. Pop, "Low Variability in Synthetic Monolayer MoS₂ Devices," *ACS Nano*, vol. 11, pp. 8456-8463, Aug 22 2017.
- [4] J. Zhang, H. Yu, W. Chen, X. Tian, D. Liu, M. Cheng, *et al.*, "Scalable growth of high-quality polycrystalline MoS(2) monolayers on SiO(2) with tunable grain sizes," *ACS Nano*, vol. 8, pp. 6024-30, Jun 24 2014.
- [5] J. B. Smith, D. Hagaman, and H. F. Ji, "Growth of 2D black phosphorus film from chemical vapor deposition," *Nanotechnology*, vol. 27, p. 215602, May 27 2016.
- [6] S. Das and J. Appenzeller, "WSe₂ field effect transistors with enhanced ambipolar characteristics," *Applied Physics Letters*, vol. 103, Sep 2 2013.
- [7] X. Zhang, F. Zhang, Y. Wang, D. S. Schulman, T. Zhang, A. Bansal, *et al.*, "Defect-Controlled Nucleation and Orientation of WSe₂ on hBN: A Route to Single-Crystal Epitaxial Monolayers," *ACS Nano*, 2019/02/13 2019.
- [8] D. S. Schulman, A. J. Arnold, and S. Das, "Contact engineering for 2D materials and devices," *Chem Soc Rev*, Mar 2 2018.
- [9] S. Chuang, C. Battaglia, A. Azcatl, S. McDonnell, J. S. Kang, X. Yin, *et al.*, "MoS₂ p-type transistors and diodes enabled by high work function MoO_x contacts," *Nano letters*, vol. 14, pp. 1337-1342, 2014.
- [10] D. J. Perello, S. H. Chae, S. Song, and Y. H. Lee, "High-performance n-type black phosphorus transistors with type control via thickness and contact-metal engineering," *Nature communications*, vol. 6, p. 7809, 2015.
- [11] Y. Wang, Z. Li, L. Feng, H. Bai, and C. Wang, "Hardware design of multiclass SVM classification for epilepsy and epileptic seizure detection," *IET Circuits, Devices & Systems*, vol. 12, pp. 108-115, 2018.
- [12] A. Saidi, S. Ben Othman, W. Kacem, and S. Ben Saoud, "FPGA implementation of EEG signal analysis system for the detection of epileptic seizure," pp. 415-420, 2018.
- [13] M. Saleheen, H. Alemzadeh, A. M. Cheriyan, Z. Kalbarczyk, and R. K. Iyer, "An efficient embedded hardware for high accuracy detection of epileptic seizures," pp. 1889-1896, 2010.
- [14] M. A. B. Altaf and W. Saadeh, "A 0.21 μ J patient-specific REM/Non-REM sleep classifier for Alzheimer patients," pp. 1-4, 2017.
- [15] Y. Abate, D. Akinwande, S. Gamage, H. Wang, M. Snure, N. Poudel, *et al.*, "Recent Progress on Stability and Passivation of Black Phosphorus," *Adv Mater*, p. e1704749, May 11 2018.
- [16] S.-L. Yau, T. P. Moffat, A. J. Bard, Z. Zhang, and M. M. Lerner, "STM of the (010) surface of orthorhombic phosphorus," *Chemical Physics Letters*, vol. 198, pp. 383-388, 1992.
- [17] S. Gamage, A. Fali, N. Aghamiri, L. Yang, P. D. Ye, and Y. Abate, "Reliable passivation of black phosphorus by thin hybrid coating," *Nanotechnology*, vol. 28, p. 265201, Jun 30 2017.
- [18] Y. Y. Illarionov, M. Waltl, G. Rzepa, J. S. Kim, S. Kim, A. Dodabalapur, *et al.*, "Long-Term Stability and Reliability of Black Phosphorus Field-Effect Transistors," *ACS Nano*, vol. 10, pp. 9543-9549, Oct 25 2016.
- [19] S. Das, M. Demarteau, and A. Roelofs, "Ambipolar phosphorene field effect transistor," *ACS nano*, vol. 8, pp. 11730-11738, 2014.

- [20] S. Das, H. Y. Chen, A. V. Penumatcha, and J. Appenzeller, "High performance multilayer MoS₂ transistors with scandium contacts," *Nano Lett*, vol. 13, pp. 100-5, Jan 09 2013.
- [21] D. L. Olynick, B. Cord, A. Schipotinin, D. F. Ogletree, and P. J. Schuck, "Electron-beam exposure mechanisms in hydrogen silsesquioxane investigated by vibrational spectroscopy and in situ electron-beam-induced desorption," *Journal of Vacuum Science & Technology B, Nanotechnology and Microelectronics: Materials, Processing, Measurement, and Phenomena*, vol. 28, pp. 581-587, 2010.
- [22] A. Wang, B. H. Calhoun, and A. P. Chandrakasan, *Sub-threshold Design for Ultra Low-Power Systems (Series on Integrated Circuits and Systems)*: Springer-Verlag, 2006.
- [23] S. Hanson, M. Seok, D. Sylvester, and D. Blaauw, "Nanometer device scaling in subthreshold logic and SRAM," *IEEE Transactions on Electron Devices*, vol. 55, pp. 175-185, 2008.
- [24] A. Wang and A. Chandrakasan, "A 180-mV subthreshold FFT processor using a minimum energy design methodology," *IEEE Journal of solid-state circuits*, vol. 40, pp. 310-319, 2005.
- [25] A. G. Andreou, K. Boahen, P. O. Pouliquen, A. Pavasovic, R. E. Jenkins, and K. Strohhahn, "Current-mode subthreshold MOS circuits for analog VLSI neural systems," *IEEE Transactions on neural networks*, vol. 2, pp. 205-213, 1991.
- [26] R. M. Wallace and G. D. Wilk, "High- κ Dielectric Materials for Microelectronics," *Critical Reviews in Solid State and Materials Sciences*, vol. 28, pp. 231-285, 2003.
- [27] S. Das, W. Zhang, M. Demarteau, A. Hoffmann, M. Dubey, and A. Roelofs, "Tunable Transport Gap in Phosphorene," *Nano Letters*, vol. 14, pp. 5733-5739, 2014/10/08 2014.
- [28] H. Fang, S. Chuang, T. C. Chang, K. Takei, T. Takahashi, and A. Javey, "High-performance single layered WSe₂ p-FETs with chemically doped contacts," *Nano Lett*, vol. 12, pp. 3788-92, Jul 11 2012.

Reviewers' comments:

Reviewer #1 (Remarks to the Author):

The manuscript was carefully revised by the authors based on the comments. Before acceptance, one important reference [“2D MoS2 Neuromorphic Devices for Brain-Like Computational Systems”. Small. 13, 1700933, (2017)] on the topic of MoS2 synaptic/neuromorphic devices should be cited.

Reviewer #2 (Remarks to the Author):

The authors have addressed my concerns and I suggest now it can be accepted.

Reviewer #3 (Remarks to the Author):

The revised version still makes the same unsubstantiated claims of “beating Boltzmann and von Neumann limits”.

The following sentence in the abstract need to be removed:

We further describe how such Gaussian synapses can defeat the Boltzmann limit to reinstate energy scaling, quantum limit to restore size scaling and von Neumann limit to facilitate complexity scaling.

The following sentence needs to be nuanced:

Finally, as an elucidation, we demonstrate seamless classification of neural oscillations also known as the brainwave patterns (δ , θ , α , β , γ) by exploiting Gaussian synapse based probabilistic neural network (PNN) architecture and by using the Electroencephalography (EEG) data as the training sample.

In particular, “demonstrate” is not correct and needs to be replaced by “show simulation results suggesting”.

The same change needs to be made to “demonstrate” in the following sentence on p. 14:

Next, we demonstrate how PNNs based on Gaussian synapses can be used for the classification of various neural oscillations, also known as the brainwaves that are fundamental to human awareness, cognition, emotions and actions.

and also “demonstrates” on p. 16:

Finally, Fig. 5a demonstrates the PNN architecture for the detection of new brainwave patterns.

and in the conclusion on p. 18:

Finally, we demonstrate that PNN architecture based on Gaussian synapses is capable of recognizing complex neural oscillations or brainwave patterns from large volumes of EEG data with extreme energy efficiency.

Other comments: do a thorough check for spelling and grammar errors, e.g.:

Delbruck is misspelled.

Remove "the" from "In the conclusion"

etc.

Response to Reviewers' Comments
Reviewer's Comment
Our Response
Changes Made in the Manuscript

Reviewer #1 (Remarks to the Author):

The manuscript was carefully revised by the authors based on the comments. Before acceptance, one important reference [“2D MoS₂ Neuromorphic Devices for Brain-Like Computational Systems”. Small. 13, 1700933, (2017)] on the topic of MoS₂ synaptic/neuromorphic devices should be cited.

We thank the reviewer for reading our revised manuscript and recommending acceptance.

We have added the new reference in the revised manuscript

Reviewer #2 (Remarks to the Author):

The authors have addressed my concerns and I suggest now it can be accepted.

We thank the reviewer for recommending acceptance of our manuscript.

Reviewer #3 (Remarks to the Author):

The revised version still makes the same unsubstantiated claims of “beating Boltzmann and von Neumann limits”.

The following sentence in the abstract need to be removed:

We further describe how such Gaussian synapses can defeat the Boltzmann limit to reinstate energy scaling, quantum limit to restore size scaling and von Neumann limit to facilitate complexity scaling.

We have removed this statement from the abstract since it does not take away any key findings reported in this manuscript.

The following sentence needs to be nuanced:

Finally, as an elucidation, we demonstrate seamless classification of neural oscillations also known as the brainwave patterns (δ , θ , α , β , γ) by exploiting Gaussian synapse based probabilistic neural network (PNN) architecture and by using the Electroencephalography (EEG) data as the training sample.

In particular, “demonstrate” is not correct and needs to be replaced by “show simulation results suggesting”.

We agree with the reviewer’s suggestion.

We have revised the abstract accordingly with much more concrete and focused demonstration of the device electronic capabilities.

The same change needs to be made to “demonstrate” in the following sentence on p. 14:

Next, we demonstrate how PNNs based on Gaussian synapses can be used for the classification of various neural oscillations, also known as the brainwaves that are fundamental to human awareness, cognition, emotions and actions.

and also “demonstrates” on p. 16:

Finally, Fig. 5a demonstrates the PNN architecture for the detection of new brainwave patterns.

and in the conclusion on p. 18:

Finally, we demonstrate that PNN architecture based on Gaussian synapses is capable of recognizing complex neural oscillations or brainwave patterns from large volumes of EEG data with extreme energy efficiency.

We agree with the reviewer’s suggestion.

We have revised the manuscript texts accordingly.

Other comments: do a thorough check for spelling and grammar errors, e.g.:

Delbruck is misspelled.

Remove “the” from “In the conclusion”

We thank the reviewer for pointing out the typos. We have now performed a thorough grammar and spell check.

We have revised the manuscript texts accordingly.